# SURFER v2.0: A flexible and simple model linking anthropogenic CO$_2$ emissions and solar radiation modification to ocean acidification and sea level rise

Marina Martínez Montero[1], Michel Crucifix[1], Victor Couplet[1], Nuria Brede[2,4], and Nicola Botta[2,3]

[1]Earth and Life Institute, UCLouvain, Louvain-la-Neuve, Belgium
[2]Potsdam Institute for Climate Impact Research, Potsdam, Germany
[3]Chalmers University of Technology, Göteborg, Sweden
[4]University of Potsdam, Potsdam, Germany

**Correspondence:** Marina Martínez Montero (marina.martinez@uclouvain.be)

**Abstract.** We present SURFER, a novel reduced model for estimating the impact of CO$_2$ emissions and solar radiation modification options on sea level rise and ocean acidification over timescales of several thousands of years. SURFER has been designed for the analysis of CO$_2$ emission and solar radiation modification policies, for supporting the computation of optimal (CO$_2$ emission and solar radiation modification) policies and for the study of commitment and responsibility under uncertainty. The model is based on a combination of conservation laws for the masses of atmospheric and oceanic carbon and for the oceanic temperature anomalies, and of ad-hoc parametrisations for the different sea level rise contributors: ice sheets, glaciers and ocean thermal expansion. It consists of 9 loosely coupled ordinary differential equations, is understandable, fast and easy to modify and calibrate. It reproduces the results of more sophisticated, high-dimensional earth system models on time scales up to millennia.

## 1 Introduction

Carbon emissions in the following decades will have a significant impact on the sea level and on the acidity of the oceans for millennia (e.g., Clark et al., 2016; Van Breedam et al., 2020), mainly because of the long residence time of CO$_2$ in the atmosphere-ocean system (Archer, 2005; Archer et al., 2009). While reducing CO$_2$ emissions is necessary, different geoengineering methods have been proposed to reduce some of the negative effects of climate change and complement the efforts of CO$_2$ emission reduction (Lawrence et al., 2018). One approach to such climate intervention is that of solar radiation modification, which proposes to reflect some of the incoming solar radiation to cool Earth's surface. Solar radiation modification, in particular stratospheric aerosol intervention, acts on much shorter timescales than the residence time of CO$_2$ in the atmosphere.

Therefore, assigning a value judgement to climate mitigation policies for the next decades necessarily has to account for impacts (e.g., in terms of benefits and damages due to sea level rise, crossing planetary boundaries (Rockström et al., 2009; Steffen et al., 2015) and perhaps solar radiation modification and CO$_2$ emission reduction costs) over much longer time scales than the next decades.

Failing to do so may lead to short sighted policies that commit upcoming generations to unmanageable impacts or that severely shrinks their range of viable $CO_2$ emission and/or solar radiation modification options (Clark et al., 2016; Mengel et al., 2018; Nauels et al., 2019).

Assessing the value of climate mitigation policies for the next decades over millennia requires accounting for epistemic (Shepherd et al., 2018; Shepherd, 2019) model uncertainties but also, and perhaps more importantly, for uncertainties about how these policies will actually be implemented: we know that decisions in matter of greenhouse gas abatements may be implemented with delays and that solar radiation modification options for mitigating the impacts of high $CO_2$ concentrations in the atmosphere are not free from risks, see (Gardiner, 2010; Moreno-Cruz and Keith, 2013; Robock, 2016, 2020; Helwegen

et al., 2019; Zarnetske et al., 2021). Also, assessing the value of climate mitigation policies for the next decades based on sea level rise and ocean acidification necessarily has to account for uncertainties about the (anthropogenic) forcings to be expected after, say, 2100.

Depending on the methodology applied and on the level of guarantees (correctness) required, computing "best" climate policies under uncertainty and estimating measures of responsibility for specific climate decisions (Botta et al., 2021) can be more

or less computationally expensive than assessing the value of a given policy under the same uncertainties. As a consequence, in presence of uncertainty, various interesting assessments require models that are easy to understand and modify, and fast to apply. Such assessments include (among others) the analysis of $CO_2$ emission policies, the computation of optimal policies, the assessment of commitment, responsibility and safe operating spaces (Heitzig et al., 2016), e.g., in terms of planetary boundaries (Rockström et al., 2009).

## 1.1 What this paper is about

This paper presents SURFER, a tool for estimating the Sea level Uprise Response under Forcings of Emissions and solar Radiation modification, that has been designed to meet all the above requirements. SURFER is a simple carbon-climate-sea level rise model based on a novel combination of conservation laws for the masses of atmospheric and oceanic carbon and for the oceanic temperature anomalies, and of ad-hoc parametrisations for the different sea level rise contributors.

It consists of 9 loosely coupled ordinary differential equations (ODEs) that are easy to understand and modify. Model simulations over 10k years (obtained with standard numerical approximations for stiff ODEs) typically run in less than 0.005 seconds on a standard laptop, see Sec. 2.5.

The model is easy to calibrate and reproduces the results of high-dimensional Earth system Models of Intermediate Complexity (EMICs) and Earth System Models (ESMs), capturing well the responses of global mean surface temperature anomaly,

atmospheric carbon concentration, sea level rise and ocean acidification to anthropogenic forcing.

The main focus of this paper is twofold: on the one hand, we discuss and motivate the model equations and parameters and contrast these equations and parameters to those found in similar reduced models and climate emulators. This is done in detail in Secs. 2 and 4. The aim of Sec. 2 is to establish full model transparency and, hopefully, understandability. Developing a tool for estimating the impacts of $CO_2$ emissions and solar radiation modification measures on sea level rise and on the acidification

of the oceans over millennia necessarily means making a number of choices and compromises. These choices and compromises

are motivated by the intended applications and in Sec. 4 we provide recommendations for extending the model of Sec. 2 to applications that go beyond those targeted by SURFER.

On the other hand, we provide numerical evidence that, for the intended applications, SURFER can indeed reproduce the results of more sophisticated, high-dimensional earth system models. This is done in Sec. 3.

## 1.2 What the paper is not about

Before turning to the specification of the model equations of SURFER, let us shortly discuss what this paper is not about.

We have argued that one intended application of SURFER is that of assessing the value of $CO_2$ emission policies for the next decades based on sea level rise and ocean acidification under different kinds of uncertainty. We do not provide an example of such an application here. Besides the specification of an emission policy, this would also require specifying

– the uncertainties that affect the specific problem at stake,

– a function for measuring the value of possible trajectories (sometimes called a metric) and, most importantly,

– a measure for aggregating probability distributions (often the expected value measure)

for the specific problem at stake. This would go well beyond the scope of this paper but we refer the reader to (Botta et al., 2018, 2021; Helwegen et al., 2019; Carlino et al., 2020) for a discussion of the steps involved in setting up a stylised problem. Similarly, we do not apply SURFER to the computation of optimal policies (Botta et al., 2018; Helwegen et al., 2019; Carlino et al., 2020) or to the study of commitment and responsibility (Martínez Montero et al., 2022; Botta et al., 2021) under uncertainty here. Instead, we will do so in a dedicated companion paper.

A final caveat is at place: SURFER is not meant to be a "better" model than the reference ESMs it is intended to emulate in terms of climate metrics, and has not been calibrated to minimize a well-defined distance from a set of "trusted" models (see Sec. 3). Rather, SURFER is a tool that trades model complexity for speed and understandability. Both properties are crucial for the intended applications: speed is needed to tackle the computational complexity associated with uncertainty; And, under uncertainty, nothing is worse than applying a tool that one does not fully understand.

## 2 Model specification

SURFER consists of a system of 9 ordinary differential equations for the masses of carbon in four different reservoirs (atmosphere, upper ocean layer, deeper ocean layer and land), the temperature anomalies of two different reservoirs (upper ocean layer and deeper ocean layer), the volume of Greenland and Antarctic ice sheets and sea level rise related to glaciers. Sea level rise due to the ocean thermal expansion is also taken into account through a parametrisation in terms of the ocean temperature anomalies.

Two external forcings drive the system: anthropogenic $CO_2$ emissions and solar radiation modification in the form of $SO_2$ injections.

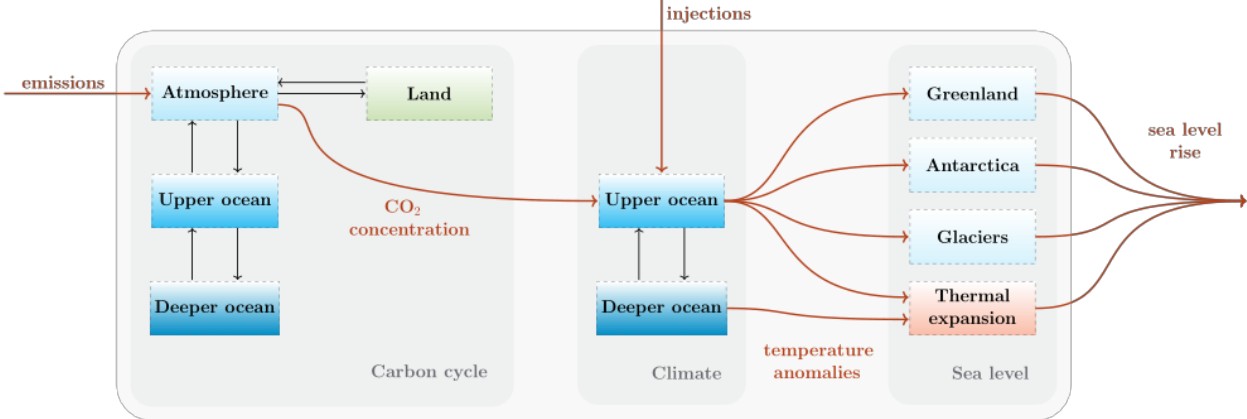

**Figure 1.** Conceptual diagram of SURFER. The state variables are indicated by the boxes, interactions and sources are depicted by black and dark orange arrows respectively.

The $CO_2$ emission rate, through the accumulation of $CO_2$ in the atmosphere and due to the greenhouse effect, leads to long-lasting temperature increases, while the $SO_2$ injections, through reflecting some of the incoming solar radiation, are responsible for fast but short-lasting temperature decreases. We refer the readers to (Moreno-Cruz and Keith, 2013; Helwegen et al., 2019; Robock, 2020) for more information on this form of geoengineering and to (Visioni et al., 2021) for the the latests results from the Geoengineering Model Inter-comparison Project. The $CO_2$ emissions are the source in the carbon cycle sub-model, which evolves the amount of carbon in the considered carbon reservoirs. The carbon concentration in the atmosphere and the $SO_2$ injections are the driving forces of the climate sub-model, which evolves the temperature anomalies in two thermal reservoirs. Finally, the temperature anomaly of the upper ocean layer (assumed in equilibrium with the atmosphere and land) forces the melting of the ice sheets and glaciers, causing sea level to rise, while both ocean layers contribute to sea level rise due to the ocean's thermal expansion. Fig. 1 provides a conceptual diagram for the model. We have not included other interactions between the sub-models but we address the most relevant feedbacks in Sec. 4.

### 2.1 Carbon cycle model

The carbon cycle model is based on BEAM, a simple carbon cycle model developed by Glotter et al. (2014) for economic and policy analyses. BEAM stands for "Bolin and Eriksson Adjusted Model" in acknowledgement of (Bolin and Eriksson, 1959). Since we modified the model by Glotter et al. (2014) by including an extra carbon reservoir, a land reservoir[1], along with minor modifications to the original equations, we proceed with a full presentation of the carbon cycle model.

---

[1]In (Bolin and Eriksson, 1959) they include hummus and vegetation reservoirs. We have followed a different approach for obtaining the equations and fluxes corresponding to the land reservoir.

The equations for the Atmosphere ($A$), upper ocean layer ($U$), deeper ocean layer ($D$) and land ($L$) carbon reservoirs read as follows

$$\frac{dM_A}{dt} = E - F_{A\to U} - F_{A\to L}, \tag{1}$$

$$\frac{dM_U}{dt} = F_{A\to U} - F_{U\to D}, \tag{2}$$

$$\frac{dM_D}{dt} = F_{U\to D}, \tag{3}$$

$$\frac{dM_L}{dt} = F_{A\to L}, \tag{4}$$

where $M_i$ is the mass of carbon in reservoir $i$, $F_{i\to j}$ the net carbon flux from reservoir $i$ to reservoir $j$ and $E$ is the anthropogenic carbon emission rate. As part of the land reservoir we consider only soils and vegetation, ignoring carbon in permafrost and fossil fuel reserves. Sinks and sources associated with carbon outgassing, weathering and sediment burial are ignored because they are of secondary importance at the timescales considered here (10 yr to 5000 yr).

### 2.1.1 $F_{A\to U}$

Modelling the carbon flux between the atmosphere and the ocean relies on fundamental ocean carbonate chemistry which we now summarise (see Ch. 8 of textbook by Sarmiento and Gruber (2006) for a deeper treatment).

When $CO_2$ in the atmosphere goes into the ocean it undergoes a series of chemical reactions

$$CO_{2(gas)} + H_2O \quad \rightleftharpoons \quad H_2CO_3^*, \tag{R1}$$

$$H_2CO_3^* \quad \rightleftharpoons \quad HCO_3^- + H^+, \tag{R2}$$

$$HCO_3^- \quad \rightleftharpoons \quad CO_3^{2-} + 2H^+, \tag{R3}$$

where $H_2CO_3^*$ represents a mix of aqueous carbon dioxide, $CO_{2(aqueous)}$, and carbonic acid, $H_2CO_3$[2]. The distribution between the three carbon species, $H_2CO_3^*$, $HCO_3^-$ (bicarbonate), and $CO_3^{2-}$ (carbonate), is fast with respect to the ocean's circulation timescale, and hence equilibrium is assumed. The equilibrium distribution relations

$$K_1 = \frac{[H^+][HCO_3^-]}{[H_2CO_3^*]}, \qquad K_2 = \frac{[H^+][CO_3^{2-}]}{[HCO_3^-]} \tag{5}$$

are dictated by the ocean's acidity, quantified by the proton concentration $[H^+]$. $K_1$ and $K_2$ are dissociation constants and $[H^+]$, measured in moles per kilogram, relates to the ocean pH as

$$pH = -\log_{10}[H^+]. \tag{6}$$

---

[2]It is common practice to consider these two species of carbon together into a single variable because they are difficult to distinguish from each other (Sarmiento and Gruber, 2006, Ch. 8.2).

The total dissolved inorganic carbon (DIC) can then be written as

$$\begin{aligned}
\text{DIC} &= [\text{H}_2\text{CO}_3^*] + [\text{HCO}_3^-] + [\text{CO}_3^{2-}] \\
&= \left(1 + \frac{K_1}{[\text{H}^+]} + \frac{K_1 K_2}{[\text{H}^+]^2}\right)[\text{H}_2\text{CO}_3^*].
\end{aligned} \tag{7}$$

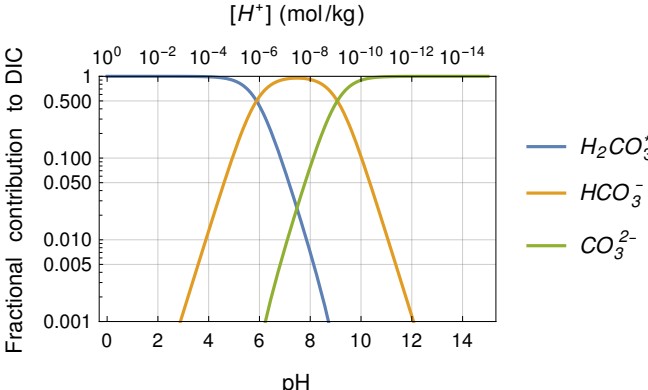

**Figure 2.** Carbon species fractional contribution to DIC for varying pH.

In Fig. 2 we show the fractional contributions of the different carbon species to DIC for varying pH. For the present pH value of around 8 we can see in Fig. 2 that bicarbonate is the dominant carbon species in the ocean. From the chemical reactions (R1), (R2) and (R3) we see that when $\text{CO}_2$ dissolves in the ocean, hydrogen ions are released and ocean acidifies. This in turn means that the proportion of carbonate decreases and that of $\text{H}_2\text{CO}_3^*$ increases. The alkalinity however, defined as the excess of bases over acids

$$Q = [\text{HCO}_3^-] + 2[\text{CO}_3^{2-}] + [\text{OH}^-] - [\text{H}^+] + [\text{B(OH)}_4^-] + \text{minor bases}, \tag{8}$$

does not change with those reactions. Since no other reactions are accounted for in this carbon cycle model the alkalinity is constant. This assumption will lead to stronger than expected acidification on long timescales ($\sim 4000$ years) in which calcium carbonate production, dissolution and burial (not accounted for here) are significant. As it is usual practice, we will approximate the alkalinity[3] by its dominant terms, that is by the carbonate alkalinity

$$Q \approx [\text{HCO}_3^-] + 2[\text{CO}_3^{2-}] = \left(\frac{K_1}{[\text{H}^+]} + \frac{2K_1 K_2}{[\text{H}^+]^2}\right)[\text{H}_2\text{CO}_3^*]. \tag{9}$$

The flow of $\text{CO}_2$ between the atmosphere and upper ocean layer is proportional to the difference in $\text{CO}_2$ partial pressures

$$F_{A \to U} \propto \left(p_{\text{CO}_2}^A - p_{\text{CO}_2}^U\right) \tag{10}$$

and by writing the partial pressures as proportional to the corresponding carbon masses

$$F_{A \to U} = -k_{A \to U} M_A + k_{U \to A} M_U' \tag{11}$$

---

[3] See e.g., Ch. 8 Sec. 8.2 of Sarmiento and Gruber (2006) for more details on this.

where $p_{CO_2}^U$ refers to the partial pressure of carbon in the form $H_2CO_3^*$, and $M_U'$ to the corresponding carbon mass. The parameters $k_{i \to j}$ are the transport coefficients from reservoir $i$ to reservoir $j$.

The equilibrium concentration of $H_2CO_3^*$ in the ocean, corresponding to an atmospheric $CO_2$ partial pressure $p_{CO_2}^A$ can be determined through the $CO_2$ solubility constant

$$K_0 = \frac{[H_2CO_3^*]}{p_{CO_2}}, \tag{12}$$

where $p_{CO_2}$ is written without a reservoir index because, when in equilibrium, atmospheric and upper ocean have the same $CO_2$ partial pressure.

### 2.1.2  $F_{U \to D}$

The exchange of carbon between the two ocean layers is ruled by oceanic currents and therefore depends on the total dissolved inorganic carbon in each layer

$$F_{U \to D} = k_{U \to D} M_U - k_{D \to U} M_D. \tag{13}$$

This is in contrast with the carbon exchange between the upper ocean to atmosphere which depends on the upper ocean carbon concentration in the form of $H_2CO_3^*$. Now we can write the carbon cycle equations as

$$\frac{dM_A}{dt} = E - k_{A \to U} M_A + k_{U \to A} M_U' - F_{A \to L}, \tag{14a}$$

$$\frac{dM_U}{dt} = k_{A \to U} M_A - k_{U \to A} M_U' - k_{U \to D} M_U + k_{D \to U} M_D, \tag{14b}$$

$$\frac{dM_D}{dt} = k_{U \to D} M_U - k_{D \to U} M_D, \tag{14c}$$

$$\frac{dM_L}{dt} = F_{A \to L}. \tag{14d}$$

Following Bolin and Eriksson (1959) we assume that the four reservoirs were in equilibrium at pre-industrial times (with $E(t_{PI}) = 0$) and we examine the equilibrium equations

$$0 = -k_{A \to U} M_A(t_{PI}) + k_{U \to A} M_U'(t_{PI}) - F_{A \to L}(t_{PI}), \tag{15a}$$

$$0 = k_{A \to U} M_A(t_{PI}) - k_{U \to A} M_U'(t_{PI}) - k_{U \to D} M_U(t_{PI}) + k_{D \to U} M_D(t_{PI}), \tag{15b}$$

$$0 = k_{U \to D} M_U(t_{PI}) - k_{D \to U} M_D(t_{PI}), \tag{15c}$$

$$0 = F_{A \to L}(t_{PI}). \tag{15d}$$

These allow to relate transport coefficients

$$k_{A \to U} = k_{U \to A} \frac{M_U'(t_{PI})}{M_A(t_{PI})}, \tag{16}$$

$$k_{U \to D} = k_{D \to U} \frac{M_D(t_{PI})}{M_U(t_{PI})}, \tag{17}$$

where we can further write

$$M'_U = [H_2CO_3^*]\, W_U\, \bar{m}_C$$

$$M_A = \text{moles of } CO_2 \text{ in atmosphere} \times \bar{m}_C,$$

$$M_{(U,D)} = \text{DIC}_{(U,D)}\, W_{(U,D)}\, \bar{m}_C,$$

where $W_U$ and $W_D$ stand for the whole mass of the upper and deeper ocean layers and can be approximated by

$$W_U \approx m_O\, \bar{m}_W\, \frac{h_U}{h_U + h_D}, \quad W_D \approx m_O\, \bar{m}_W\, \frac{h_D}{h_U + h_D}, \tag{18}$$

with $m_O$ the moles of water in the ocean, $\bar{m}_W$ the molar mass of $H_2O$, $\bar{m}_C$ the molar mass of carbon and $h_U$ and $h_D$ the thicknesses of the ocean layers. Considering the equilibrium solubility relation (12) and

$$p_{CO_2}^A = 1(\text{atm})\frac{\text{moles of } CO_2 \text{ in atmosphere}}{\text{moles in atmosphere}}$$

we arrive at

$$k_{A \to U} = k_{U \to A}\frac{M'_U(t_{PI})}{M_A(t_{PI})} = k_{U \to A}\frac{W_U\, K_0}{m_A}, \tag{19}$$

$$k_{U \to D} = k_{D \to U}\frac{M_D(t_{PI})}{M_U(t_{PI})} = k_{D \to U}\, \delta_{\text{DIC}}\frac{h_D}{h_U}, \tag{20}$$

where $m_A$ are the moles of air in the atmosphere and

$$\delta_{\text{DIC}} = \frac{\text{DIC}_D(t_{PI})}{\text{DIC}_U(t_{PI})} \tag{21}$$

where $\text{DIC}_U(t_{PI})$ and $\text{DIC}_D(t_{PI})$ are the pre-industrial DIC concentration in upper and lower ocean layers. The parameter $\delta_{\text{DIC}}$ specifies the DIC gradient between the two ocean layers and effectively accounts for the biological and carbonate pumps.

The next step is to express the carbon mass in $H_2CO_3^*$ form, $M'_U$, in Eqs. (14) as a function of the state variables. We begin by

$$\frac{M'_U}{M_U} = \frac{[H_2CO_3^*]_U}{\text{DIC}_U} = \left(1 + \frac{K_1}{[H^+]_U} + \frac{K_1 K_2}{[H^+]^2_U}\right)^{-1}.$$

Using the definitions of DIC and carbonate alkalinity ($Q$ in Eq. (9)), and the relation of $\text{DIC}_i$ with carbon mass $M_i$, $[H^+]_i$ can be solved for in terms of $M_i$

$$[H^+]_i = \frac{K_1}{2\tilde{Q}_i}\left(\sqrt{(M_i - \tilde{Q}_i)^2 - 4\frac{K_2}{K_1}\tilde{Q}_i(\tilde{Q}_i - 2M_i)} + (M_i - \tilde{Q}_i)\right) \tag{22}$$

with $i = U$ or $D$ and where

$$\tilde{Q}_i = Q_i\, W_i\, \bar{m}_C$$

is the carbon mass corresponding to the carbonate alkalinity. The factor tracking the ocean's buffer capacity becomes

$$B(M_U) \equiv \frac{M'_U}{M_U} = \frac{1}{2} - \frac{\tilde{Q}_U}{2M_U} + \frac{K_1\sqrt{(M_U - \tilde{Q}_U)^2 - 4\frac{K_2}{K_1}\tilde{Q}_U(\tilde{Q}_U - 2M_U)} - 4M_U K_2}{2M_U(K_1 - 4K_2)}. \tag{23}$$

### 2.1.3  $F_{A \to L}$

Now we turn to the equation for the land reservoir. The proposed equation is not process based, it is instead based on the output of the Zero Emissions Commitment Model Inter-comparison Project (ZECMIP) (Jones et al., 2019; MacDougall et al., 2020) experiments by different EMICs and ESMs.

We analysed the output of the ZECMIP experiments B1 and B3 in (Jones et al., 2019; MacDougall et al., 2020), and observed that the land carbon anomaly relaxes to a value proportional to the atmospheric carbon anomaly after typically 4 to 6 decades. This behaviour can be captured by the following relation:

$$\delta M_L(t) = \alpha_L \delta M_A(t), \tag{24}$$

where $\delta M_i(t) = M_i(t) - M_i(t_{PI})$. The values of $\alpha_L$ approached by the different models in B1 experiments (lower cumulated emissions) tend to be higher than the ones approached by the higher cumulated emissions experiment B3. We noticed that the quantity

$$\frac{\delta M_L(t)}{\delta M_A(t)} \frac{M_A(t)}{M_A(t_{PI})} \tag{25}$$

tends to approach a model-dependent constant value, say $\beta_L$, which is independent of the cumulated emissions after four to six decades. Based on these observations we propose the following equation for the land carbon anomaly $\delta M_L$

$$\frac{d\delta M_L}{dt} = k_{A \to L} \left( \beta_L \frac{M_A(t_{PI})}{M_A} \delta M_A - \delta M_L \right). \tag{26}$$

With this equation $\delta M_L$ relaxes to an equilibrium value proportional to the ratio $\frac{\delta M_A}{M_A}$. This dependency can be interpreted as the result of two competing processes : CO2 fertilization ($\propto \delta M_A$) and an enhanced bacterial respiration due to climate change ($\propto M_A$)

The final equations for the carbon cycle can now be written as

$$\frac{dM_A}{dt} = E(t) - k_{A \to U} \left( M_A - \frac{m_A}{W_U K_0} B(M_U) M_U \right) - k_{A \to L} \left( \beta_L M_A(t_{PI}) \left( 1 - \frac{M_A(t_{PI})}{M_A} \right) - (M_L - M_L(t_{PI})) \right), \tag{27a}$$

$$\frac{dM_U}{dt} = k_{A \to U} \left( M_A - \frac{m_A}{W_U K_0} B(M_U) M_U \right) - k_{U \to D} \left( M_U - \frac{1}{\delta_{\text{DIC}}} \frac{h_U}{h_D} M_D \right), \tag{27b}$$

$$\frac{dM_D}{dt} = k_{U \to D} \left( M_U - \frac{1}{\delta_{\text{DIC}}} \frac{h_U}{h_D} M_D \right), \tag{27c}$$

$$\frac{dM_L}{dt} = k_{A \to L} \left( \beta_L M_A(t_{PI}) \left( 1 - \frac{M_A(t_{PI})}{M_A} \right) - (M_L - M_L(t_{PI})) \right). \tag{27d}$$

Equations (27) are very similar to the ones presented by Glotter et al. (2014) but with the following important differences:

1. There is a land carbon reservoir. This update to the Glotter et al. (2014) model improves the agreement with most recent results from EMICs and ESMs.

2. The ocean buffer factor is explicitly written in terms of $M_U$ to highlight the non-linear nature of the model.

3. The relation between the transport coefficients between the two ocean layers depends not only on the ratio of the thickness of the layers ($\delta$) but also on the ratio of their pre-industrial concentration of dissolved inorganic carbon ($\delta_{DIC}$). This allows for an equilibrium solution in which the dissolved inorganic carbon concentration is different in the upper and lower layers, which is known to be the case due to the soft tissue and carbonate pumps.

The presented carbon cycle equations for atmosphere and ocean (and also the ones in (Glotter et al., 2014)) are similar to the ones considered in DICE, the Dynamic Integrated model of Climate and the Economy (Nordhaus, 1992, 2013). The big difference between the two is that the upper ocean buffer factor $B$ is considered to be constant in DICE while in SURFER it evolves with the ocean acidification. By including the non-linearities due to ocean carbonate chemistry, SURFER's carbon cycle, as the one by Glotter et al. (2014), captures the fact that as the ocean takes in $CO_2$ from the atmosphere it becomes a worse sink for future $CO_2$ intake. This enables the correct tracking of carbon concentrations up to timescales of several thousands of years, which is impossible with a linear model like DICE. One of the main objectives of the present contribution is to provide a model of sea level rise caused by ice sheet melting which is a slow process lasting several thousands of years. As explained by Archer et al. (2009), a linear carbon cycle is inadequate for such long-term purposes.

Another benefit of SURFER's carbon cycle is the tracking of the pH and the concentrations of the different carbon species in the ocean; this can be done a posteriori, using the obtained $M_U(t)$ together with Eqs. (22), (6), (9) and (5). SURFER can thus be of use for policy analyses that deal with ocean acidification.

Ocean acidification destabilises marine ecosystems making it more difficult for shellfish and corals to grow. As such, ocean acidification is one of the 9 identified planetary boundaries (Rockström et al., 2009; Steffen et al., 2015). This planetary boundary has however not been quantified in terms of pH. It is instead given in terms of the aragonite saturation state which can be approximated as

$$\Omega_{ar} \approx \frac{[CO_3^{2-}]}{[CO_3^{2-}]_{saturation\ ar}}, \tag{28}$$

where $[CO_3^{2-}]$ is the carbonate concentration in the ocean and $[CO_3^{2-}]_{saturation\ ar}$ is the carbonate concentration at which aragonite is saturated[4]. Aragonite is the most vulnerable form of calcium carbonate and its saturation state relates in a more straightforward way to some forms of marine life than pH. A value of $\Omega_{ar} < 1$ means that aragonite dissolves but organisms struggle to grow and thrive already at bigger values of $\Omega_{ar}$. The planetary boundary is set at 80% of the pre-industrial value which was $\Omega_{ar}(t_{PI}) = 3.44$.

---

[4]$[CO_3^{2-}]_{saturation\ ar}$ depends strongly on pressure and hence changes along the water column.

## 2.2  Climate model with solar radiation modification

SURFER's climate sub-model is a linear 2-box model, similar to those in (Gregory, 2000; Held et al., 2010), for the evolution of the upper and deeper ocean temperature anomalies $\delta T_U$ and $\delta T_D$, respectively (measured in $^\circ$C)

$$c_{vol} h_U \frac{d\delta T_U}{dt} = \beta \left( \frac{F(M_A, I)}{\beta} - \delta T_U \right) - \gamma \left( \delta T_U - \delta T_D \right), \tag{29}$$

$$c_{vol} h_D \frac{d\delta T_D}{dt} = \gamma \left( \delta T_U - \delta T_D \right). \tag{30}$$

The atmosphere is assumed to be in thermal equilibrium with the upper ocean layer. Due to this, and that the upper ocean heat capacity is much bigger than that of the atmosphere, the atmosphere is not explicitly part of the climate sub-model and the radiative forcings are applied to the upper ocean layer. The constant $c_{vol}$ is the seawater's volumetric heat capacity, obtained by multiplying the specific heat capacity of seawater and the density of seawater. The thicknesses $h_U$ and $h_D$ of the two ocean layers are the same as for the carbon cycle. $\gamma$ is thermal conductivity between the ocean layers and $\beta$ is the climate feedback parameter related to the equilibrium climate sensitivity (in $^\circ$C)

$$\text{ECS} = \frac{F_{2X}}{\beta}, \tag{31}$$

with $F_{2X}$ the radiative forcing corresponding to a doubling of $CO_2$. The anthropogenic radiative forcing (measured in W m$^{-2}$) in Eq. (29) responsible for the temperature anomalies consists of two terms,

$$F(M_A, I) = F_{2X} \log_2 \left( \frac{M_A}{M_A(t_{PI})} \right) - \alpha_{SO_2} \exp \left( -(\beta_{SO_2}/I)^{\gamma_{SO_2}} \right), \tag{32}$$

a first one corresponding to the standard greenhouse effect and a second one corresponding to solar radiation modification in the form of $SO_2$ injections. The solar radiation modification term comes from (Niemeier and Timmreck, 2015). The variable $I$ corresponds to the sulfur injection rate and is in general time dependent. $\alpha_{SO_2}$, $\beta_{SO_2}$ and $\gamma_{SO_2}$ are the fitting parameters considered in (Niemeier and Timmreck, 2015).

## 2.3  Sea level rise

Four different components contribute to SURFER's estimation of sea level rise: ocean thermal expansion, glaciers and two ice sheets

$$S_{tot} = S_{th} + S_{gl} + S_{GIS} + S_{AIS}, \tag{33}$$

where $S_{tot}$ is the total sea level rise and $S_{th}$, $S_{gl}$, $S_{GIS}$ and $S_{AIS}$ are the ocean thermal expansion, glacier, Greenland ice sheet and Antarctic ice sheet contributions, respectively. While ocean thermal expansion and glaciers are the first contributors to sea level rise on the short timescales of decades, on the longer timescales of centuries and millennia, the biggest contributions come from the ice sheets.

### 2.3.1 Ocean thermal expansion

The ocean thermal expansion parametrisation relies on the ocean data (layers sizes and temperature anomalies) that is part of SURFER's climate sub-model. This contribution to sea level rise is then computed as

$$S_{th} = \alpha_U h_U \delta T_U + \alpha_D h_D \delta T_D \tag{34}$$

where $\alpha_i$ is the thermal expansion coefficient corresponding to the $i$ layer (in °C$^{-1}$), $h_i$ is the size of the $i$ layer (in meters) and $\delta T_i$ is the temperature anomaly (in °C) with respect to to pre-industrial times of the $i$ layer. We consider that the expansion coefficients of the two layers have different values to capture the fact that surface waters have bigger thermal expansion coefficients than deeper denser waters as shown, for example, in Fig. 1(c) of (Williams et al., 2012). As a simplification, we neglect the size change of the ocean layers $h_{(U,D)}$ due to sea level rise and we also assume that the expansion coefficients are constant in time.

The sea level rise contribution from ocean thermal expansion comes from both ocean layers. In the timescales of decades and a couple of centuries, the deeper ocean layer does not contribute much due to its thermal inertia. As the deeper ocean warms up, in the timescale of thousands of years, it can become the main contributor to $S_{th}$. Figure 3a shows the sea level rise commitment from ocean expansion once thermal equilibrium has been achieved between the two ocean layers, that is, with $\delta T_U = \delta T_D$.

### 2.3.2 Glaciers

The sea level rise contribution from glaciers is modelled with an ordinary differential equation that relaxes the current sea level rise value due to glaciers, $S_{gl}$, to its expected equilibrium value for the current temperature $S_{gl\,eq}(\delta T_U)$

$$\frac{dS_{gl}}{dt} = \frac{1}{\tau_{gl}}\big(S_{gl\,eq}(\delta T_U) - S_{gl}\big), \tag{35}$$

where $\tau_{gl}$ is the relaxation timescale. The same form of equation was used by Mengel et al. (2016), although with a differently parametrised $S_{gl\,eq}$. Levermann et al. (2013) analysed the sea level commitments of different sea level rise components depending on the forcing temperature. They estimate the shape of such $S_{gl\,eq}(\delta T_U)$ for all land glaciers excluding ice sheets (see Fig. 1B in (Levermann et al., 2013)) which we will approximate as

$$S_{gl\,eq}(\delta T_U) = S_{gl\,pot} \tanh\left(\frac{\delta T_U}{\zeta}\right), \tag{36}$$

where $S_{gl\,pot}$ is the potential sea level rise due to glaciers, which corresponds to all the ice volume in glaciers in units of sea level rise equivalent, and $\zeta$ is a sensitivity coefficient. Figure 3b shows the shape of $S_{gl\,eq}$ for the suggested values of $S_{gl\,pot}$ and $\zeta$ in Table 6.

This way of modelling the glaciers has a couple of advantages to similar methods used in other simple models. First of all, the formulation is fully transparent and nothing more than equations (35) and (36) are needed. As for the behavior, it captures some expected physics:

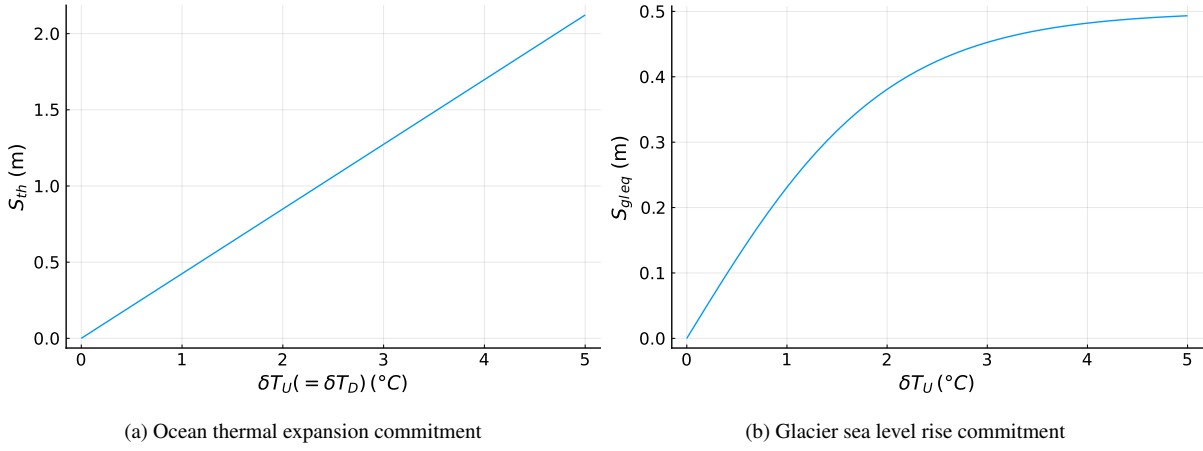

(a) Ocean thermal expansion commitment          (b) Glacier sea level rise commitment

**Figure 3.** Equilibrium sea level rise contribution from ocean thermal expansion and glaciers.

1. No forcing corresponds to no sea level rise from glaciers.

2. For small enough forcings, different levels of forcing lead to different levels of sea level rise.

3. Different levels of forcing on the same initial state generate different rates of sea level rise.

4. There is a cap on the maximum amount of sea level rise that can come from glaciers.

### 2.3.3 Ice sheets

Multi-stability regions and tipping points have been identified both for the Greenland and Antarctic ice sheets (e.g., Lenton et al., 2008; Letreguilly et al., 1991; Pattyn, 2006; Ridley et al., 2010; Robinson et al., 2012; Gregory et al., 2020; Garbe et al., 2020). The proposed ice sheet model highlights those tipping points and is easy to adapt to both ice sheets such that it captures their dynamics. The state variable is the volume fraction of ice with respect to a reference state, which we set to be the ice sheet's pre-industrial state. The ice sheets' contributions sea level rise with respect to pre-industrial times is computed as a function of the ice sheets' melted fractions of ice and their total sea level rise potential

$$S_{GIS} = S_{GIS\,pot}(1 - V_{GIS}(t)), \quad S_{AIS} = S_{AIS\,pot}(1 - V_{AIS}(t)), \tag{37}$$

where $S_{GIS\,pot}$ and $S_{AIS\,pot}$ are the sea level rise potentials of Greenland's and Antarctic ice sheet respectively, and $V_{GIS}$ and $V_{AIS}$ their volume fraction with respect to their pre-industrial volume.

To capture the dynamics of an ice sheet featuring multi-stability and tipping points we propose a non linear ordinary differential equation for the ice volume fraction

$$\frac{dV}{dt} = \mu(V, \delta T_U)\left(-V^3 + a_2 V^2 + a_1 V + c_1 \delta T_U + c_0\right), \tag{38}$$

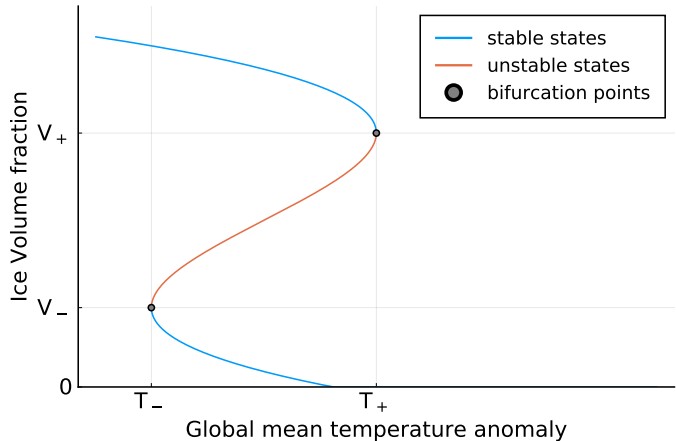

**Figure 4.** Ice sheet steady states for model defined in eqs. (38) and (39).

where the third-order polynomial of the volume fraction and the term proportional to a forcing temperature imply a double
fold bifurcation diagram for the steady states in terms of a constant forcing temperature, see Fig. 4. In contrast to the ocean
part of the carbon cycle model and climate model presented before, the ice sheet model is not explicitly derived from physical
processes; it is a generic dynamical system model based on the concept of a double-fold bifurcation to be calibrated on state-
of-the-art ice sheet models' output. In that sense, it is an emulator. The different terms of the polynomial are not there because
they represent specific physical processes but because as a whole they produce the desired steady-state structure. The constant
parameters $(a_2, a_1, c_1, c_0)$ are given in terms of the bifurcation points $((T_+, V_+), (T_-, V_-))$ as

$$a_2 = \frac{3(V_- + V_+)}{2}, \tag{39a}$$

$$a_1 = -3V_-V_+, \tag{39b}$$

$$c_1 = -\frac{(V_+ - V_-)^3}{2(T_+ - T_-)}, \tag{39c}$$

$$c_0 = +\frac{T_+V_-^2(V_- - 3V_+) - T_-V_+^2(V_+ - 3V_-)}{2(T_- - T_+)}, \tag{39d}$$

which are the quantities which determine the steady-state structure of the system, see Fig. 4. Since we want to impose the
additional constraint of there existing a steady-state ice volume fraction of 1 at temperature anomaly equal to 0, the number of
free parameters is reduced by one by setting

$$V_- = \frac{-2 + V_+\left(1 + G^{1/3} + G^{-1/3}\right)}{-1 + G^{1/3} + G^{-1/3}} \tag{40}$$

with

$$G = \left(\frac{T_+ + T_- + 2\sqrt{T_-T_+}}{T_+ - T_-}\right). \tag{41}$$

Instead of fixing the 4 parameters $((T_+, V_+), (T_-, V_-))$ independently, $V_-$ is given in terms of the other three such that the pre-industrial reference state condition $V_{eq}(\delta T_U = 0) = 1$ is satisfied.

The evolution is further affected by the forcing temperature $\delta T_U$ and inverse timescale

$$
\mu(V, \delta T_U) = \begin{cases} 1/\tau_+ & \text{if } H > 0, \\ 1/\tau_- & \text{if } H < 0 \text{ and } V > 0, \\ 0 & \text{if } H < 0 \text{ and } V = 0, \end{cases} \tag{42}
$$

with

$$
H = \left( -V^3 + a_2 V^2 + a_1 V + c_1 \delta T_U + c_0 \right). \tag{43}
$$

We write $\mu(V, \delta T_U)$ as a function of the state variables $V$ and $\delta T_U$ in such a way that it can take 3 different constant values. This is done to reflect the timescale asymmetry between the processes of melting and freezing ice and to ensure that the ice fraction remains bigger or equal to zero.

## 2.4 Calibration and initial conditions

In this section we give the values and units of the parameters used to run the model. We also provide an explanation of how we have fixed some of them and obtained the pre-industrial initial conditions that we use in Sec. 3.

Parameters and initial conditions for the carbon cycle model can be found in Tables 1 and 2, the parameters for the climate model are in Table 3 and the ones for sea level rise can be found in Tables 4, 5 and 6 for the ocean thermal expansion, glaciers and ice sheet (adapted to Greenland and Antarctica) models respectively. All examples in this paper start from pre-industrial conditions, which for the climate and sea level rise models are trivial, $\delta T_U(t_{PI}) = \delta T_D(t_{PI}) = 0$, $S_{gl}(t_{PI}) = 0$, $V_{GIS}(t_{PI}) = V_{AIS}(t_{PI}) = 1$.

### 2.4.1 Carbon cycle

The dissociation parameters $K_1$ and $K_2$, the $CO_2$ solubility $K_0$, and the alkalinity $Q$ are assumed to be constant. A temperature ($T$) and salinity ($S$) dependent alternative is provided for $K_1$, $K_2$ and $K_0$ in (Sarmiento and Gruber, 2006, Pg. 325)[5]. The specific values that we will be using for the constants $K_1$, $K_2$ and $K_0$ have been obtained from the expressions in (Sarmiento and Gruber, 2006). We fix the parameter $\delta_{\text{DIC}}$ to 1.15 in accordance with data provided in (Sarmiento and Gruber, 2006, Pg. 320).

We fix the parameters $K_1$, $K_2$ and $K_0$ in the carbon cycle model by ensuring that initial conditions are consistent with pre-industrial data. The parameter $\delta = h_D/h_U$, also playing a role in the initial conditions, has been set to $\delta = 20$, which is a middle ground between the thermal box sizes of (Gregory, 2000; Held et al., 2010) and the ratio of sizes suggested in (Bolin and Eriksson, 1959; Glotter et al., 2014), the reason for the chosen value is provided below. Other parameters, like $k_{A \to U}$,

---

[5]Similar expressions are given in (Glotter et al., 2014)

$k_{U \to D}$ and $k_{A \to L}$ have been adjusted to match the dynamics of more complex carbon cycle models. We now proceed with a more in depth explanation.

In pre-industrial times the atmospheric $CO_2$ concentration was 280ppm which corresponds to an atmospheric carbon mass

$$M_A(t_{PI}) = \text{moles of } CO_2 \text{ in atmosphere}(t_{PI}) \times \bar{m}_C$$
$$= 280 \times 10^{-6} m_A \bar{m}_C = 580.3 \text{ PgC}.$$

where $\bar{m}_C$ is the Carbon molar mass. We assume that in pre-industrial conditions the carbon cycle was in equilibrium. Additionally we impose that the total mass of carbon in the ocean was 38000 PgC. Using the relations between ocean carbon
masses, the corresponding DICs and the equilibrium equations we can write

$$M_U(t_{PI}) + M_D(t_{PI}) = \text{DIC}_U(t_{PI}) m_O \bar{m}_C \bar{m}_W \frac{1 + \delta_{\text{DIC}} \delta}{1 + \delta}$$
$$= 38000 \text{ PgC} \tag{44}$$

where we have written the carbon and water molar masses as $\bar{m}_{(C,W)}$. We see that $\delta = 20$ implies a reasonable value of $\text{DIC}_U(t_{PI}) = 1973.53 \, \mu\text{mol kg}^{-1}$ by comparing to global average data in Fig. 8.1.2 of (Sarmiento and Gruber, 2006) and
to global averages from ZECMIP, see pre-industrial range of values in Fig. 9c. This value of DIC is one of the ingredients used to fix the dissociation and solubility constants.

The pH of the upper ocean in pre-industrial times was around 8.2. Here we will fix it to 8.17 in accordance with historical CMIP6 results in (Gutiérrez et al., 2021) which implies a Hydrogen ion concentration of $[H^+](t_{PI}) = 10^{-8.17} \text{ mol kg}^{-1}$. $Q_U$ and $\text{DIC}_U$ at pre-industrial time can be written as

$$Q_U = \left( \frac{K_1}{[H^+](t_{PI})} + \frac{2K_1 K_2}{([H^+](t_{PI}))^2} \right) K_0 p_{CO_2}^A(t_{PI}),$$

$$\text{DIC}_U(t_{PI}) = \left( 1 + \frac{K_1}{[H^+](t_{PI})} + \frac{K_1 K_2}{([H^+](t_{PI}))^2} \right) K_0 p_{CO_2}^A(t_{PI})$$

where $p_{CO_2}^A(t_{PI}) = 280 \times 10^{-6}$ atm. We use these relations to fix the dissociation and solubility constants. First, we impose an alkalinity value compatible with observations $Q_U = 2200 \, \mu\text{mol kg}^{-1}$. Second, we impose the already fixed value of $\text{DIC}_U(t_{PI}) = 1973.53 \, \mu\text{mol kg}^{-1}$. Third, we use the temperature $(T)$ and salinity $(S)$ dependent expressions for the dissocia-
tion and solubility given in (Sarmiento and Gruber, 2006). Last, we solve the system of two equations for $T$ and $S$ numerically. Such a procedure yields "effective" $T = 294.7$K and $S = 32.49\%o$ a warmer and slightly less salty ocean than the global averages, but they determine dissociation constants which, in the end, yield realistic carbon masses, concentrations, and alkalinity in the pre-industrial ocean.

We ignore the temperature dependence of the carbonate concentration corresponding to aragonite saturation and we fix it to

$$[CO_3^{2-}]_{\text{saturation ar}} = \frac{[CO_3^{2-}](t_{PI})}{\Omega_{\text{ar}}(t_{PI})} \tag{45}$$

where $[CO_3^{-2}](t_{PI})$ is obtained through Eqs. (5) and (12) and $\Omega_{ar}(t_{PI}) = 3.44$ from (Rockström et al., 2009; Steffen et al., 2015).

The ZECMIP B1 and B3 experiments' outputs suggest a land carbon parameter $\beta_L$ between 0.5 and 2.3 We settle for 1.7 but the choice is not critical. This value is closer to those of ESMs than to those of the EMICs as can be seen in Figs. 9f and 10f. The pre-industrial mass of land carbon is set to 2200 PgC for plotting purposes but this quantity does not affect the land carbon uptake.

Finally, for the inverse timescale $k_{A \to U}$ we take the value recommended by Glotter et al. (2014). $k_{U \to D}$ is fixed to obtain a timescale for the deep ocean dynamics of 1000 years $k_{U \to D} = \delta \, \delta_{DIC}/1000$. The inverse timescale $k_{A \to L}$ is fixed to match the output of ZECMIP B1 and B3 experiments.

Users of the model are invited to explore other possibilities of fixing parameters but in this paper we will restrict ourselves to using SURFER only with the supplied parameters and initial conditions and we show that, despite its simplicity, its predictions are in excellent agreement with more complex models.

| Parameter | Value |
|---|---|
| $\delta$ | 20 |
| $\delta_{DIC}$ | 1.15 |
| $K_0$ | $3.148 \times 10^{-2}$ mol (kg atm)$^{-1}$ |
| $K_1$ | $1.326 \times 10^{-6}$ mol kg$^{-1}$ |
| $K_2$ | $9.198 \times 10^{-10}$ mol kg$^{-1}$ |
| $m_A$ | $1.727 \times 10^{20}$ mol |
| $m_O$ | $7.8 \times 10^{22}$ mol |
| $\bar{m}_C$ | $12 \times 10^{-3}$ kg mol$^{-1}$ |
| $\bar{m}_W$ | $18 \times 10^{-3}$ kg mol$^{-1}$ |
| $Q_U$ | $2.2 \times 10^{-3}$ mol kg$^{-1}$ |
| $\tilde{Q}_U$ | 1765.0 PgC |
| $[CO_3^{2-}]_{\text{saturation ar}}$ | $68.40 \, \mu$mol kg$^{-1}$ |
| $k_{A \to U}$ | 0.25 yr$^{-1}$ |
| $k_{U \to D}$ | 0.023 yr$^{-1}$ |
| $k_{A \to L}$ | 0.025 yr$^{-1}$ |
| $\beta_L$ | 1.7 |

**Table 1.** Parameter values for the carbon cycle model.

| Quantity | Value |
|---|---|
| $M_A(t_{PI})$ | 580.3 PgC |
| $M_U(t_{PI})$ | 1583.3 PgC |
| $M_D(t_{PI})$ | 36416.7 PgC |
| $M_L(t_{PI})$ | 2200 PgC |

**Table 2.** Initial equilibrium pre-industrial conditions obtained for the parameters specified in Table 1.

### 2.4.2 Climate

In the climate model we need to fix the parameters $h_U$ (or $h_D$, since the ratio $\delta$ has already been fixed), $c_{vol}$, $F_{2X}$, $\beta$, $\gamma$, and the parameters from the aerosol forcing $\alpha_{SO_2}, \beta_{SO_2}, \gamma_{SO_2}$. The values adopted in the following are listed on Table 3.

The sea water volumetric heat capacity $c_{vol}$ is obtained by multiplying the specific heat capacity of seawater, which is taken to be 3850 J kg$^{-1}$ °C$^{-1}$ and the density of seawater, taken to be 1027 kg m$^{-3}$. The extra radiative forcing due to a doubling concentration of $CO_2$, $F_{2X}$, has been chosen according to the IPCC 6th Assessment Report (Arias et al., 2021) and parameter $h_U$ according to (Gregory, 2000). $\beta$ and $\gamma$ have been fixed to yield Equilibrium Climate Sensitivity and Transient Climate Response

$$\text{ECS} = \frac{F_{2X}}{\beta} = 3.5 \,°\text{C}, \quad \text{TCR} = \frac{F_{2X}}{\beta + \gamma} = 2.0 \,°\text{C}, \tag{46}$$

compatible with results in the 6th IPCC Assessment Report (Arias et al., 2021), i.e., ECS very likely 2 to 5 °C and TCR very likely 1.4 to 2.2 °C . Aerosol forcing parameters come from the work of Niemeier and Timmreck (2015).

### 2.4.3 Ocean thermal expansion

The thermal expansion coefficients $\alpha_U$ and $\alpha_D$ in Eq. (34) have been first estimated by looking at the thermal expansion coefficient profile along the water column presented in Fig. 1(c) of (Williams et al., 2012) and taking into account the sizes of $h_U$ and $h_D$ in SURFER. Then they have been slightly corrected to better match the long-term trends presented by Van Breedam et al. (2020). Figures 5 and 16b show the performance of SURFER's thermal expansion parametrisation against other models both on short and long timescales.

### 2.4.4 Glaciers

We have fixed the total sea level rise potential of glaciers (since pre-industrial times) to 0.5m. This is an intermediate value between those reported by Levermann et al. (2013) and Farinotti et al. (2019). The sensitivity $\zeta$ has been fixed to 2 °C to mimic the glaciers' commitment curve in (Levermann et al., 2013). Finally, the timescale has been fixed to 200 yr which corresponds

| Parameter | Value |
|:---:|:---|
| $F_{2X}$ | 3.9 W m$^{-2}$ |
| $\beta$ | 1.1143 W m$^{-2}$ °C$^{-1}$ |
| $\gamma$ | 0.8357 W m$^{-2}$ °C$^{-1}$ |
| $h_U$ | 150 m |
| $h_D$ | 3000 m |
| $c_{vol}$ | 0.13 W yr m$^{-3}$ °C$^{-1}$ |
| $\alpha_{SO_2}$ | 65 W m$^{-2}$ |
| $\beta_{SO_2}$ | 2246 TgS yr$^{-1}$ |
| $\gamma_{SO_2}$ | 0.23 |

**Table 3.** Parameter values for the temperature module.

| Parameter | Value |
|:---:|:---|
| $\alpha_U$ | $2.3 \times 10^{-4}$ °C$^{-1}$ |
| $\alpha_D$ | $1.3 \times 10^{-4}$ °C$^{-1}$ |

**Table 4.** Thermal expansion coefficients.

to an intermediate value for the range found by Mengel et al. (2016). Figures 6 and 16c show the performance of SURFER's
glaciers model against other models both on short and long timescales.

| Parameter | Value |
|:---:|:---|
| $S_{gl\,pot}$ | 0.5 m |
| $\zeta$ | 2 °C |
| $\tau_{gl}$ | 200 yr |

**Table 5.** Glacier model parameters.

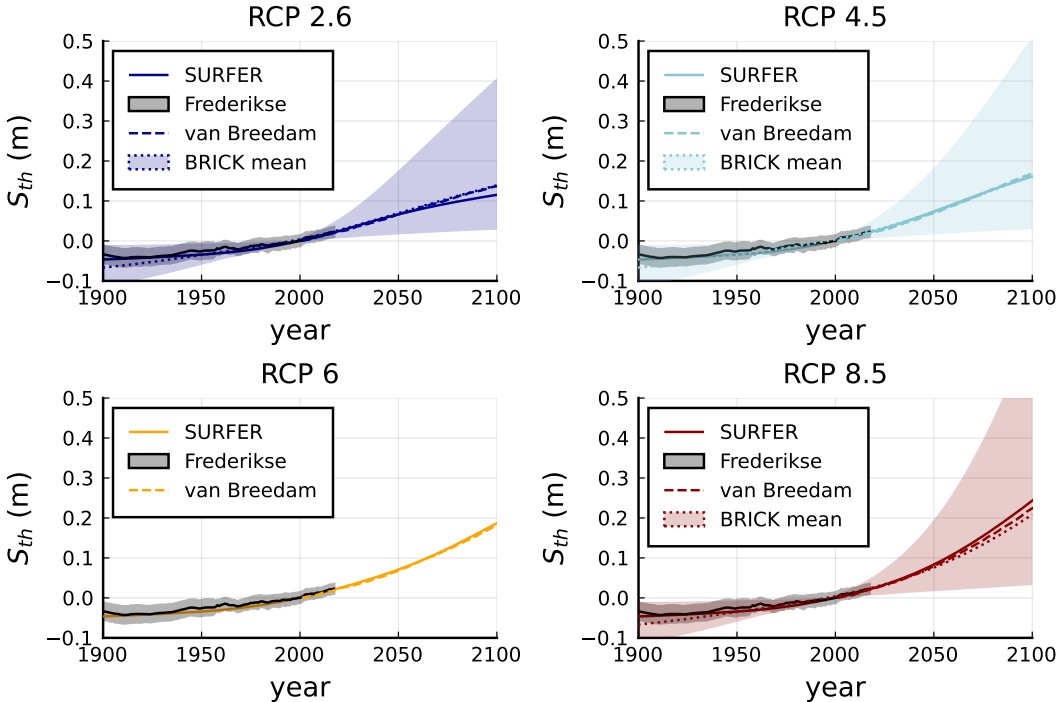

**Figure 5.** Sea level rise contribution from ocean thermal expansion for historical period and projections along RCP scenarios. SURFER's estimates correspond to the solid coloured lines, observations for the historical period from (Frederikse et al., 2020) are shown with solid black lines with the corresponding uncertainty range in grey shade. BRICK's mean and uncertainty range are shown with dotted lines and coloured shades. The dashed lines correspond to results of Van Breedam et al. (2020). Sea level rise is with respect to year 2000.

### 2.4.5 Ice sheets

We now proceed to fix the values of the parameters $(T_+, T_-, V_+, \tau_+, \tau_-)$[6] for adapting the ice sheet model to Greenland and Antarctica.

For Greenland, we have calibrated the bifurcation points $(T_\pm, V_+)$ by requiring the steady states of Eq. (38) to reproduce part of the steady state structure found in (Robinson et al., 2012). In Fig. 7a we show the upper and lower steady state branches found by Robinson et al. (2012) together with SURFER's double-fold steady state structure. Then, the melting timescale $\tau_-$ has been fixed to match the constant forcing transient results of Robinson et al. (2012)[7], see Fig. 7b. Finally, not many references present accumulation (ice sheet growth) experiments which are needed to fix $\tau_+$. For this reason we used different references to fix $\tau_+$ (Letreguilly et al., 1991; Pattyn, 2006). The calibration of $\tau_+$ was done in the way as that of $\tau_-$, i.e., seeking to reproduce the dynamic accumulation experiments of in (Letreguilly et al., 1991; Pattyn, 2006).

---

[6]The parameter $V_-$ has been fixed by using Eq. (40).

[7]Notice that $\delta T_U$ is the global mean temperature anomaly and that their plots are for regional summer temperature anomaly. On their supplementary information they explain how to relate the two.

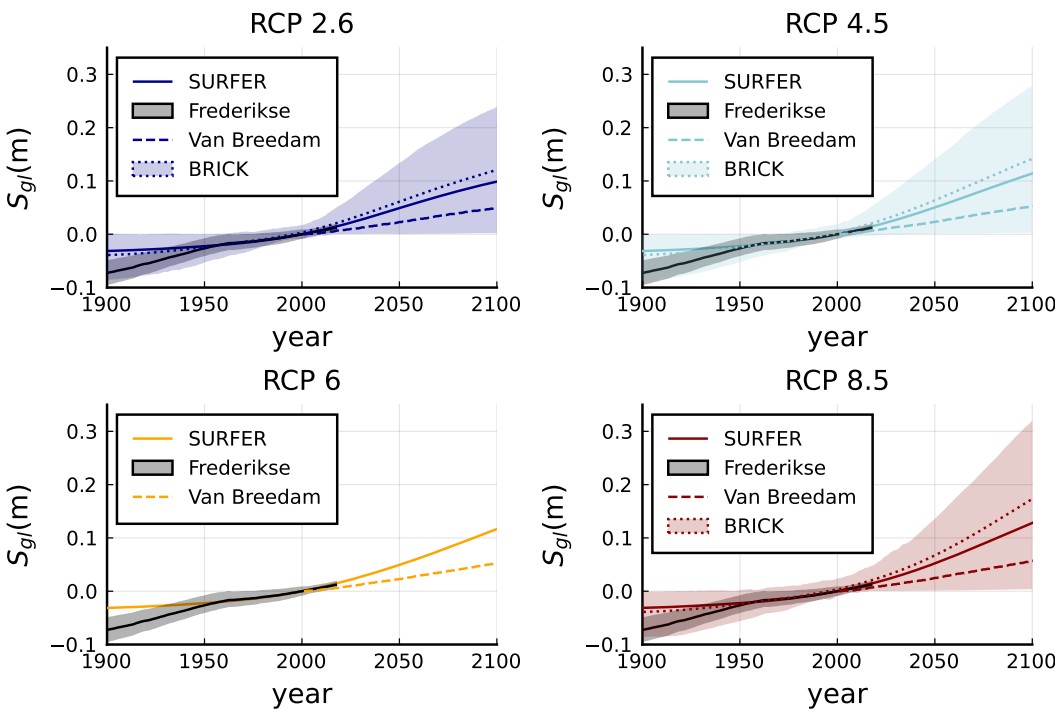

**Figure 6.** Sea level rise contribution from glaciers for historical period and projections along RCP scenarios. SURFER's estimates correspond to the solid coloured lines, observations for the historical period from (Frederikse et al., 2020) are shown with solid black lines with the corresponding uncertainty range in grey shade. BRICK's mean and uncertainty range are shown with dotted lines and coloured shades. The dashed lines correspond to results of Van Breedam et al. (2020). Sea level rise is with respect to year 2000.

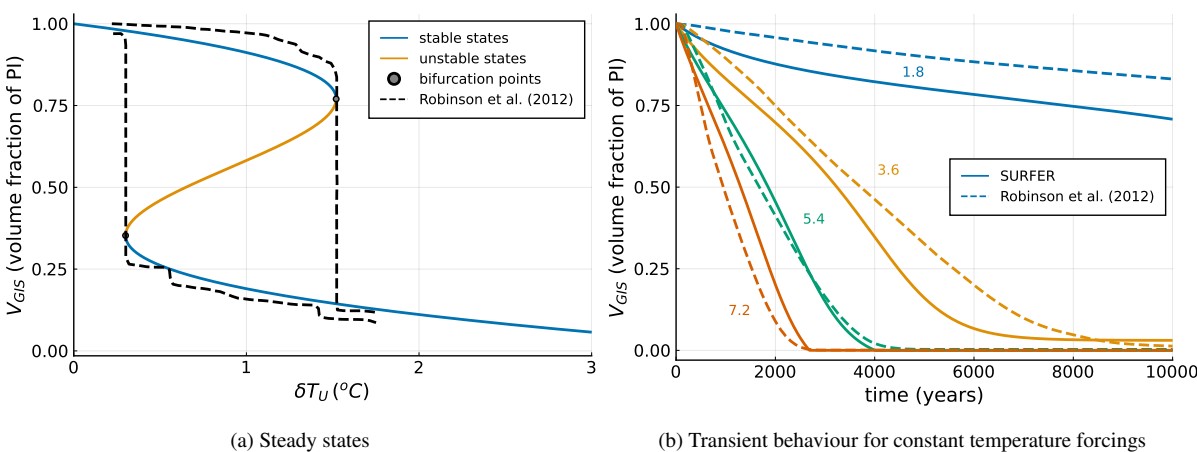

(a) Steady states          (b) Transient behaviour for constant temperature forcings

**Figure 7.** SURFER's Greenland ice sheet against the work of Robinson et al. (2012).

For the case of Antarctica fitting was more complicated than for Greenland. Contrary to the previous case, we are not aware of results from the same Antarctic ice sheet model setup for both steady state and transient experiments. For example, for the PISM model, Garbe et al. (2020) provided the steady state structure and Winkelmann et al. (2015) performed some transient experiments but since the climate models used for the forcing were not the same, the results are not completely compatible[8].

With this less than ideal situation, we have relied on the two most recent references, (Garbe et al., 2020; Van Breedam et al., 2020), to fix SURFER's Antactica parameters $(T_\pm, V_+, \tau_-)$. The steady state experiments of Garbe et al. (2020) have oriented us towards the range of acceptable parameters values while we have finally fixed those values by attempting to fit the dynamical experiments of Van Breedam et al. (2020). Again, in the absence of time-series results from accumulation experiments in the same references, $\tau_+$ has been fixed by fitting to the results of (Pattyn, 2006; Huybrechts, 1993). Fig. 16e contrasts SURFER's

prediction for $S_{AIS}$ when forced with extended RCP scenarios to the results of Van Breedam et al. (2020).

Finally, Greenland's and Antarctica's ice sheet' sea level rise potential, linking the ice volume fraction to an eustatic sea level rise, come from (Van Breedam et al., 2020) and (Garbe et al., 2020) respectively.

As more models provide both steady state structure and transient experiments, fitting can be improved by following the same strategy as used for Greenland. Additionally, it would be best if results from bigger models were presented separately for East

and West Antarctica since the two components have different tipping points and evolve with different characteristic timescales. Then SURFER could treat East and West Antarctica as two separate ice sheets.

Again, SURFER is intended to be tuned on state-of-the art models. Its parameters can, therefore, be revised as new simulations become available. The best scenario would be to have a model inter-comparison project on the timescales of millennia for both ice sheets.

| Parameter | Greenland's value | Antarctica's value |
|-----------|-------------------|--------------------|
| $T_+$ | 1.52 °C | 6.8 °C |
| $T_-$ | 0.3 °C | 4.0 °C |
| $V_+$ | 0.77 | 0.44 |
| $V_-$ | 0.3527 | 0.079 |
| $\tau_+$ | 5500 yr | 5500 yr |
| $\tau_-$ | 470 yr | 3000 yr |
| $S_{pot}$ | 7.4 m | 55 m |

**Table 6.** Parameter values used here for Greenland and Antarctic ice sheets.

---

[8]More melting happens in the setup of Winkelmann et al. (2015) than it would be expected by the results in (Garbe et al., 2020).

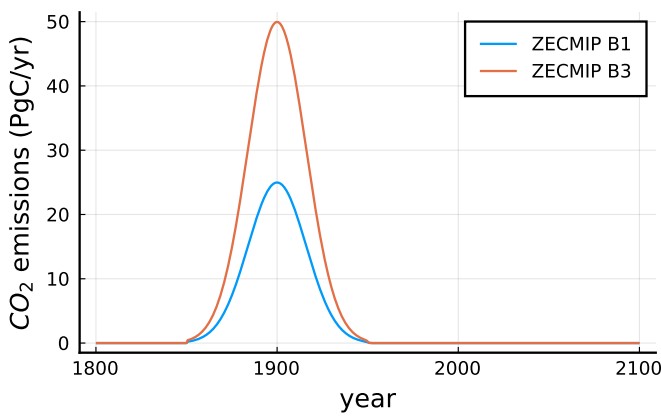

**Figure 8.** $CO_2$ emission curves for ZECMIP experiments B1 and B3.

 ## 2.5 Numerics

The model has been implemented in The Julia Programming Language (Bezanson et al., 2017) in the jupyter-lab environment. The equations have been integrated using the package `DifferentialEquations.jl` with the integration method `Rosenbrock23()` with `abstol=1e-12` and `reltol=1e-3`. The model runs extremely fast, each run of up to 10000 years taking $\approx 0.003$ seconds on a laptop with processor Intel® Core™ i7-9850H CPU @ 2.60GHz × 12.

 # 3 Numerical results and comparisons

In this section we show the model's behaviour when forced by different $CO_2$ emission scenarios and $SO_2$ injections. Only the example in Sec. 3.5 considers $SO_2$ injections; all other examples are forced with $CO_2$ emissions alone. The different examples are meant to showcase the different parts of the model. Section 3.1 illustrates the long term behaviour of the carbon cycle components, Sec. 3.2 the different ocean acidification metrics that are derivable with SURFER, Sec. 3.3 and Sec. 3.4 focus  on the short and long term sea level rise behaviour respectively and finally Sec. 3.5 deals with solar radiation modification. Whenever it has been possible to retrieve outputs of other models or historical data we have done so for making the comparisons easier. All examples start from pre-industrial conditions, more details on initial conditions can be found in Sec. 2.4.

## 3.1 ZECMIP B1 and B3 experiments

The Zero Emissions Commitment Model Intercomparison Project (ZECMIP) (Jones et al., 2019; MacDougall et al., 2020) was  proposed to quantify the amount of unrealized temperature change that occurs after $CO_2$ emissions cease and to investigate the geophysical drivers behind this climate response. The ZECMIP B1 and B3 experiments consist in starting with pre-industrial conditions and forcing the system with the bell shaped emission curves in Fig. 8, corresponding to 1000 PgC cumulated emissions for the B1 experiment and 2000 PgC for the B3 experiment. We consider the output of 6 EMICs and 2 ESMs

that participated in the ZECMIP experiment. The EMICs are Bern3D-LPX-ECS3K, DCESS1.0, MESM, MIROC-lite-LCM, PLASIM-GENIE, UVicESCM2.10 and the ESMs are GFDL-ESM2M and NorESM2-LM. Figures 9 and 10 show the good agreement of SURFER's output to that of the ZECMIP experiments. SURFER's land model is closer to the ESMs than the EMICs although this can be changed by recalibrating $\beta_L$ on the carbon cycle equations.

## 3.2 Ocean acidification under RCP or SSP forcing scenarios

In this section we run SURFER forced with historic $CO_2$ emissions followed by $CO_2$ emissions associated to the Representative Concentration Pathways (RCPs) and Shared Socio-economic Pathways (SSPs) together with their extensions (Meinshausen et al., 2011, 2020)[9], see Fig. 11. For the RCPs emission scenarios we have considered, as Van Breedam et al. (2020), that $CO_2$ emissions are zero from 2300 onwards. For the SSPs emission scenarios, as in (Meinshausen et al., 2020), we consider that $CO_2$ emissions are zero from 2250.

Figure 12 shows the evolution of different variables when forced by these scenarios. As $CO_2$ is emitted into the atmosphere, the $CO_2$ concentration in the atmosphere rises and so does the temperature and the DIC in the ocean's upper layer. Together with the increase in DIC, the pH, carbonate concentration and aragonite saturation state decrease. For the aragonite saturation state $(\Omega_{ar})_U$, in Fig. 12e, we have shadowed the region beyond the planetary boundary for ocean acidification. SURFER predicts that all considered emission scenarios cross that planetary boundary. Scenarios RCP8.5, SSP5-8.5 and SSP3-7 reach values of aragonite saturation state smaller or close to 1; in those scenarios aragonite would dissolve in the upper ocean.

Figure 13 compares SURFER's pH prediction to that of CMIP5 and CMIP6 shown in (Kwiatkowski et al., 2020). SURFER exhibits more acidification than CMIP5 and CMIP6 but it is in better agreement with CMIP6 runs.

## 3.3 Historical sea level rise and centennial projections

In this section we again run SURFER forced with historic $CO_2$ emissions followed by $CO_2$ emissions associated to the Shared Socio-economic Pathways (SSPs) together with their extensions (Meinshausen et al., 2020) up to year 2150.

We then contrast, in Fig. 14, SURFER's sea level rise predictions to the historic observations reported in (Frederikse et al., 2020) and the projections reported in Table 9.9 of IPCC AR6 WGI, see (Fox-Kemper et al., 2021). The agreement of SURFER's predictions with both data sets is remarkable given the simplicity of the model.

## 3.4 Long-term (millennial) sea level rise projections

Here we show the model's long-term behaviour when forced by historic anthropogenic $CO_2$ emissions followed by the extended RCPs emission scenarios, see (Moss et al., 2010; Meinshausen et al., 2011), which we continue to extend from the year 2300 onwards with zero emissions, as in (Van Breedam et al., 2020), see Fig. 11.

---

[9]For the SSPs we were only able to find the emission data up to 2100. We modelled the extension with a linear function going to zero by 2250. This is what was done by Meinshausen et al. (2020) except for the lowest emission scenario SSP1-2.6.

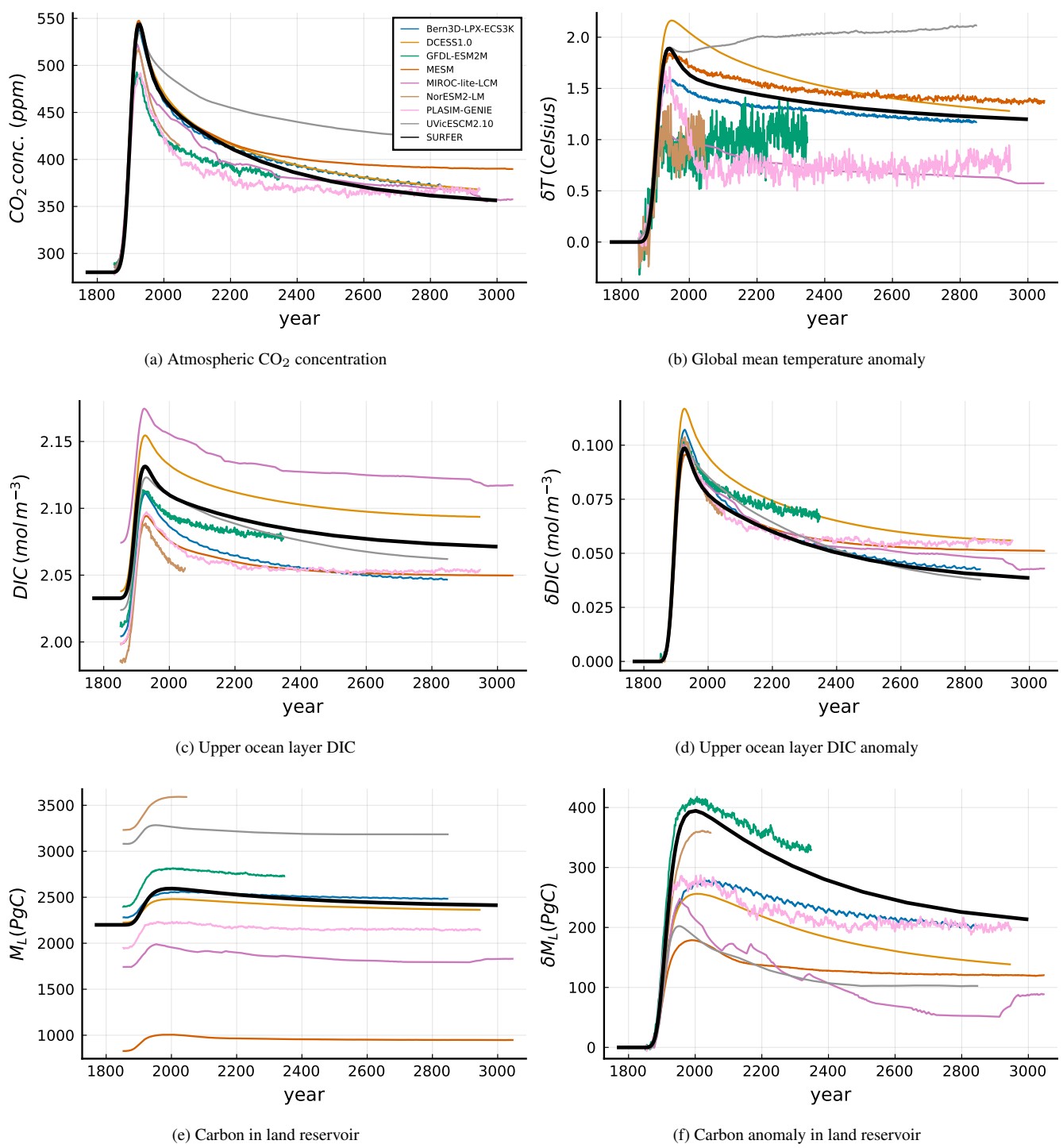

(a) Atmospheric CO$_2$ concentration

(b) Global mean temperature anomaly

(c) Upper ocean layer DIC

(d) Upper ocean layer DIC anomaly

(e) Carbon in land reservoir

(f) Carbon anomaly in land reservoir

**Figure 9.** ZECMIP B1 experiment outputs of 6 EMICs, 2 ESMs and SURFER.

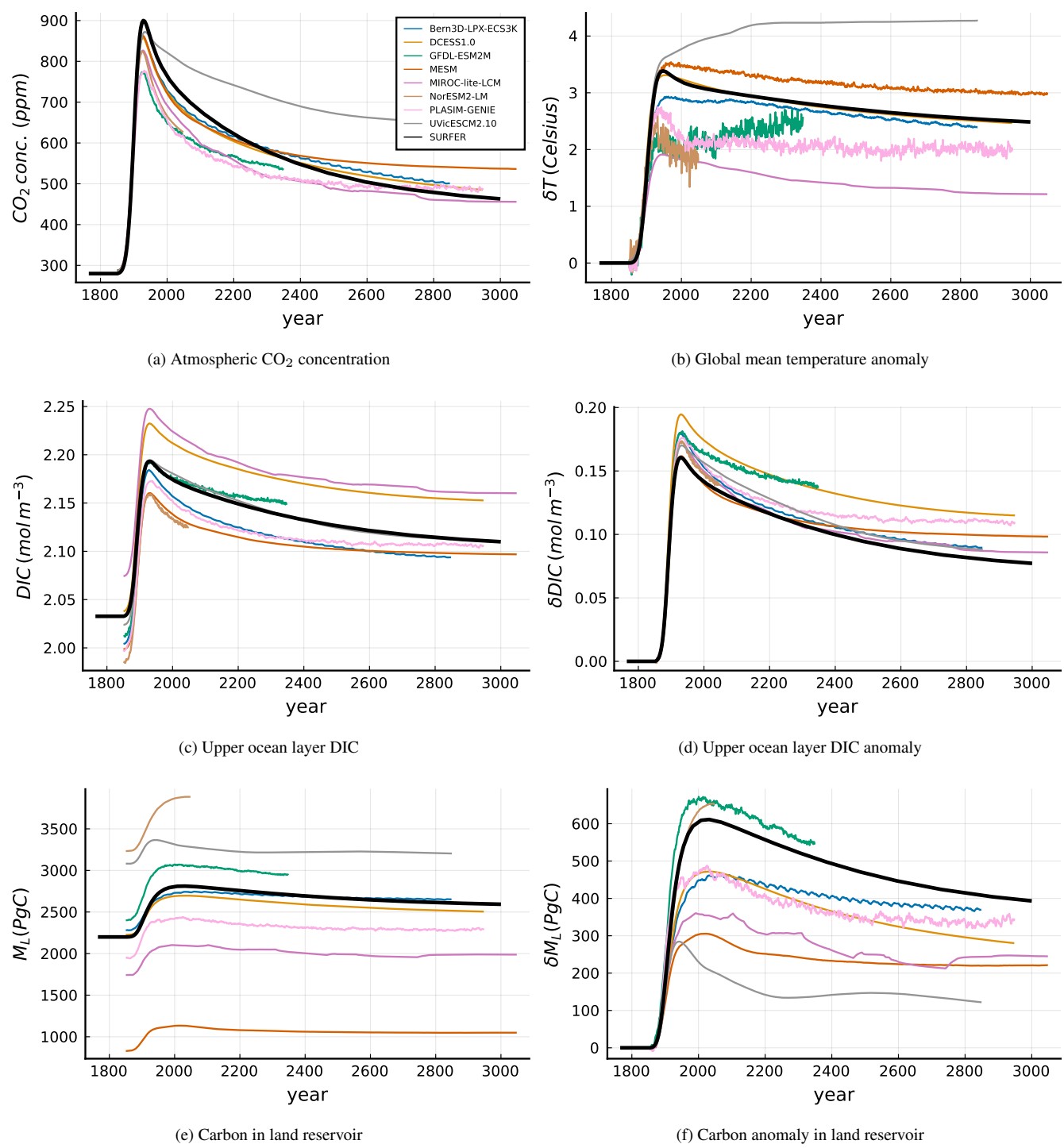

(a) Atmospheric CO$_2$ concentration

(b) Global mean temperature anomaly

(c) Upper ocean layer DIC

(d) Upper ocean layer DIC anomaly

(e) Carbon in land reservoir

(f) Carbon anomaly in land reservoir

**Figure 10.** ZECMIP B3 experiment outputs of 6 EMICs, 2 ESMs and SURFER.

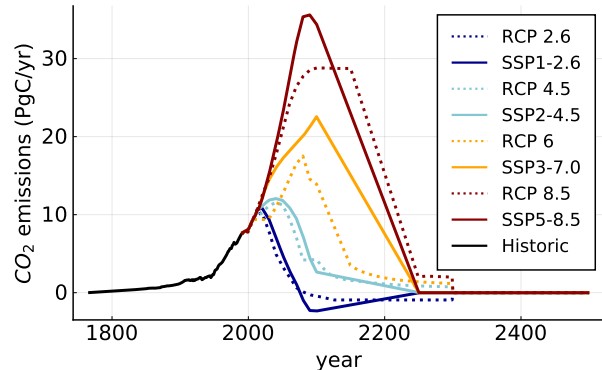

**Figure 11.** CO$_2$ emission rate for the RCP and SSP scenarios with their extensions.

Figure 15 shows the atmospheric CO$_2$ concentration and the global mean temperature anomaly for a period of 10,000 years and Fig. 16 contrasts SURFER's sea level rise predictions for each of the considered contributors with the corresponding results of Van Breedam et al. (2020).

Due the absence of calcium carbonate reactions in the carbon sub-model and the fact that SURFER does not include any feedback from the ice sheet melting on the temperature, quantities from the carbon cycle and climate sub-models reach equilibrium around year 6,000, see Fig. 15. After that time only the ice sheet components of the model continue to evolve. If carbonate compensation was included, a slow lowering of the atmospheric CO$_2$ concentration and temperature would be observed, together with its corresponding effect on the ice sheets[10]. Albedo feedbacks of the ice sheets on the climate sub-model would act

in the opposite direction increasing the temperature. These effects, as mentioned in Sec. 4, are expected to be relatively small.

    The agreement of SURFER's total sea level rise with the results of Van Breedam et al. (2020) is good for the higher emission scenarios and worse for the lower ones, see Fig. 16. The main source for the discrepancies comes, unsurprisingly, from the Greenland ice sheet' contribution. The reason for this is that we calibrated Greenland's parameters by fitting to the results of Robinson et al. (2012) and that the tipping points do not coincide: Greenland's ice sheet in (Robinson et al., 2012), and hence

SURFER's, seems to have a higher temperature tipping point than the one of in (Van Breedam et al., 2020). The agreement for Greenland's ice sheet with (Van Breedam et al., 2020), see Fig. 16d, is therefore not good for the lower emission scenarios because the corresponding forcing temperatures lie close to this threshold in SURFER but are clearly beyond it in (Van Breedam et al., 2020).

    SURFER's ocean thermal expansion parametrisation yields a $S_{th}$ that agrees well with the results of (Van Breedam et al.,

2020), see Fig 16b, similarly for the Antarctic ice sheet, see Fig 16e. For the glaciers we have different behaviour although the same order of magnitude which, as expected, is very sub-leading with respect to the other contributors at these timescales. While for SURFER the different RCP scenarios lead to different contributions from the glaciers (higher emission scenarios

---

[10]We expect this effect to be small because most of the ice sheet melting occurs in the first thousands of years where carbonate compensation still plays a very subdominant role.

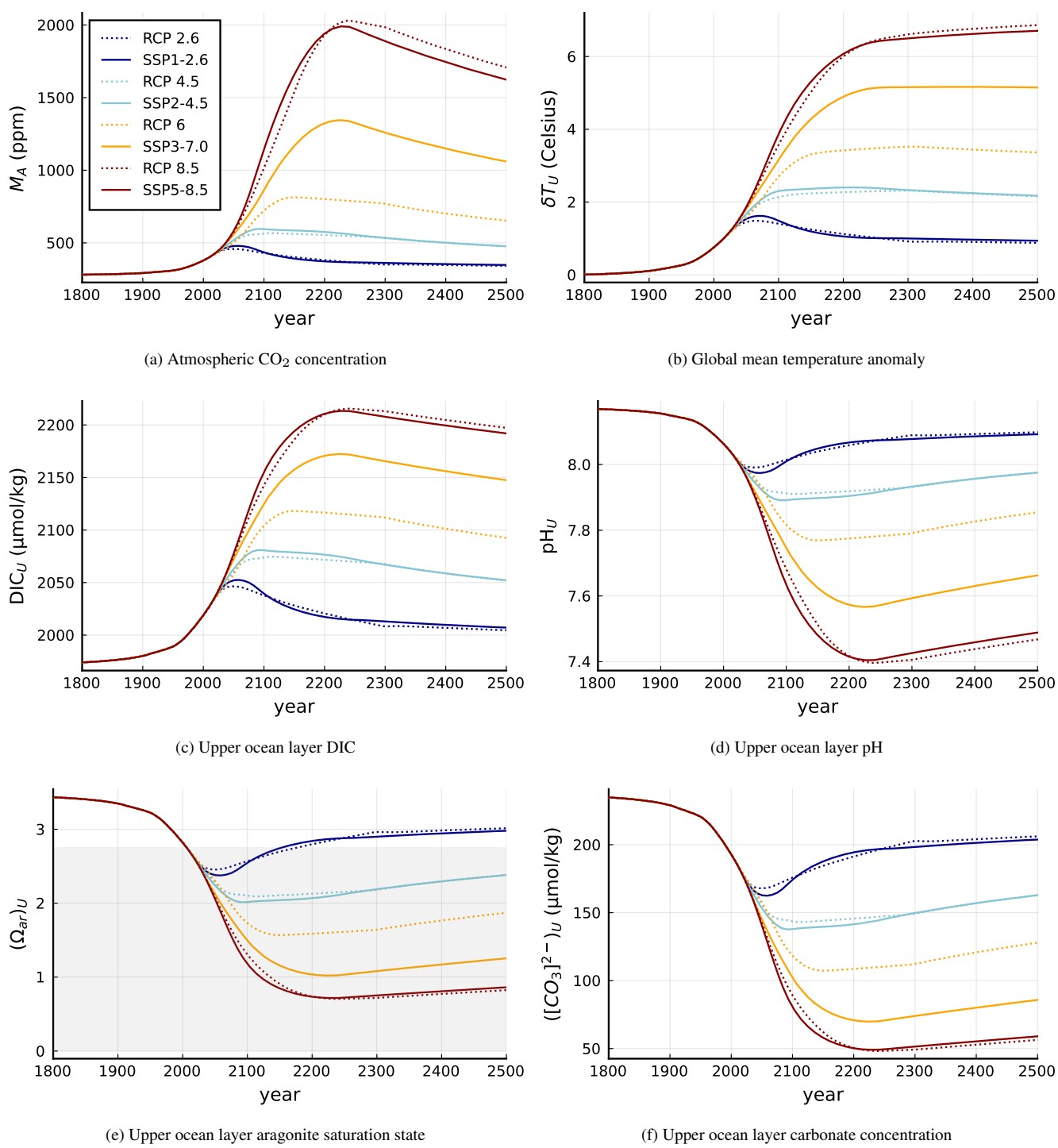

(a) Atmospheric CO$_2$ concentration

(b) Global mean temperature anomaly

(c) Upper ocean layer DIC

(d) Upper ocean layer pH

(e) Upper ocean layer aragonite saturation state

(f) Upper ocean layer carbonate concentration

**Figure 12.** SURFER's output when forced by the extended RCPs and SSPs emission scenarios.

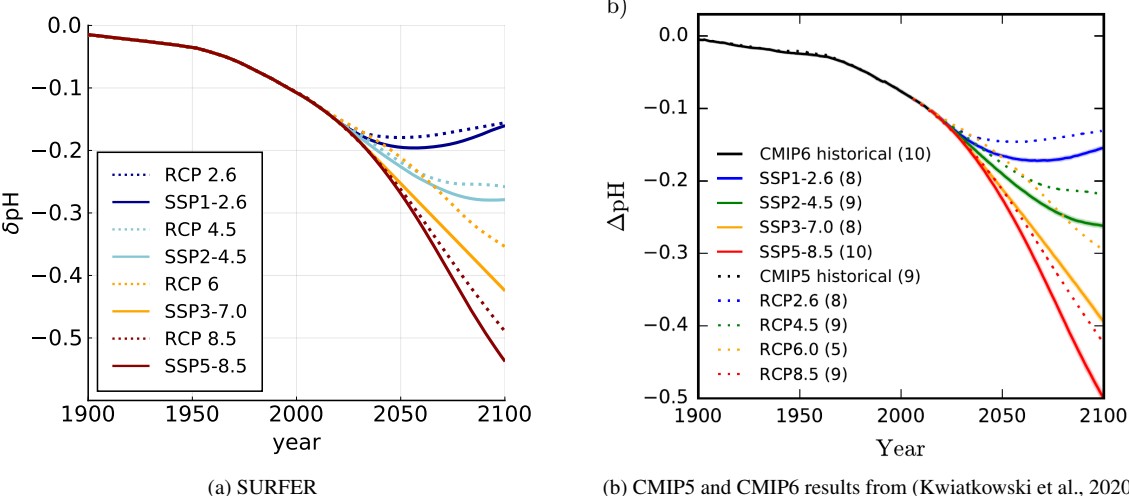

(a) SURFER

(b) CMIP5 and CMIP6 results from (Kwiatkowski et al., 2020)

**Figure 13.** Change in pH in the upper ocean layer under RCPs and SSPs emission scenarios.

leading to more melting), in (Van Breedam et al., 2020) all the considered ice in glaciers eventually melts at a rate that is scenario-dependent.

## 3.5 Solar radiation modification

As a last example we perform an experiment done by the Geoengineering Model Inter-comparison Project (GeoMIP) and compare SURFER's output to the results of Visioni et al. (2021). We focus on the G6sulfur experiment introduced in (Kravitz et al., 2015). In the G6sulfur experiment, stratospheric aerosols are injected into the model with the goal of reducing the magnitude of the net anthropogenic radiative forcing from a high forcing scenario (SSP5-8.5) to match that of a medium forcing scenario (SSP2-4.5).

We run SURFER forced by the $CO_2$ emissions corresponding to the SSP scenarios SSP5-8.5 and SSP2-4.5 in the absence of solar radiation modification. We compute the radiative forcing difference between the two scenarios, and we perform a third run in which SURFER is forced by the $CO_2$ emissions corresponding to scenario SSP5-8.5 and with sulfur injections exactly compensating the extra radiative forcing in SSP5-8.5 with respect to SSP2-4.5

$$I(t) = \beta_{SO_2} \left( -\log \left( \frac{F^{(\text{SSP5-8.5})}(t) - F^{(\text{SSP2-4.5})}(t)}{\alpha_{SO_2}} \right) \right)^{1/\gamma_{SO_2}} \tag{47}$$

with $F^{(\text{SSP5-8.5})}$ and $F^{(\text{SSP2-4.5})}$ the time varying radiative forcing obtained under the corresponding scenarios.

Figure 17 shows SURFER predictions for $CO_2$ concentration, radiative forcing, temperature and pH of upper ocean layer. This shows that while the radiative forcing, and hence the temperature, is lowered by the injection of aerosols, $CO_2$ concentration is high and therefore the ocean is dangerously acidic. Notice that this behaviour appears in SURFER by construction: in

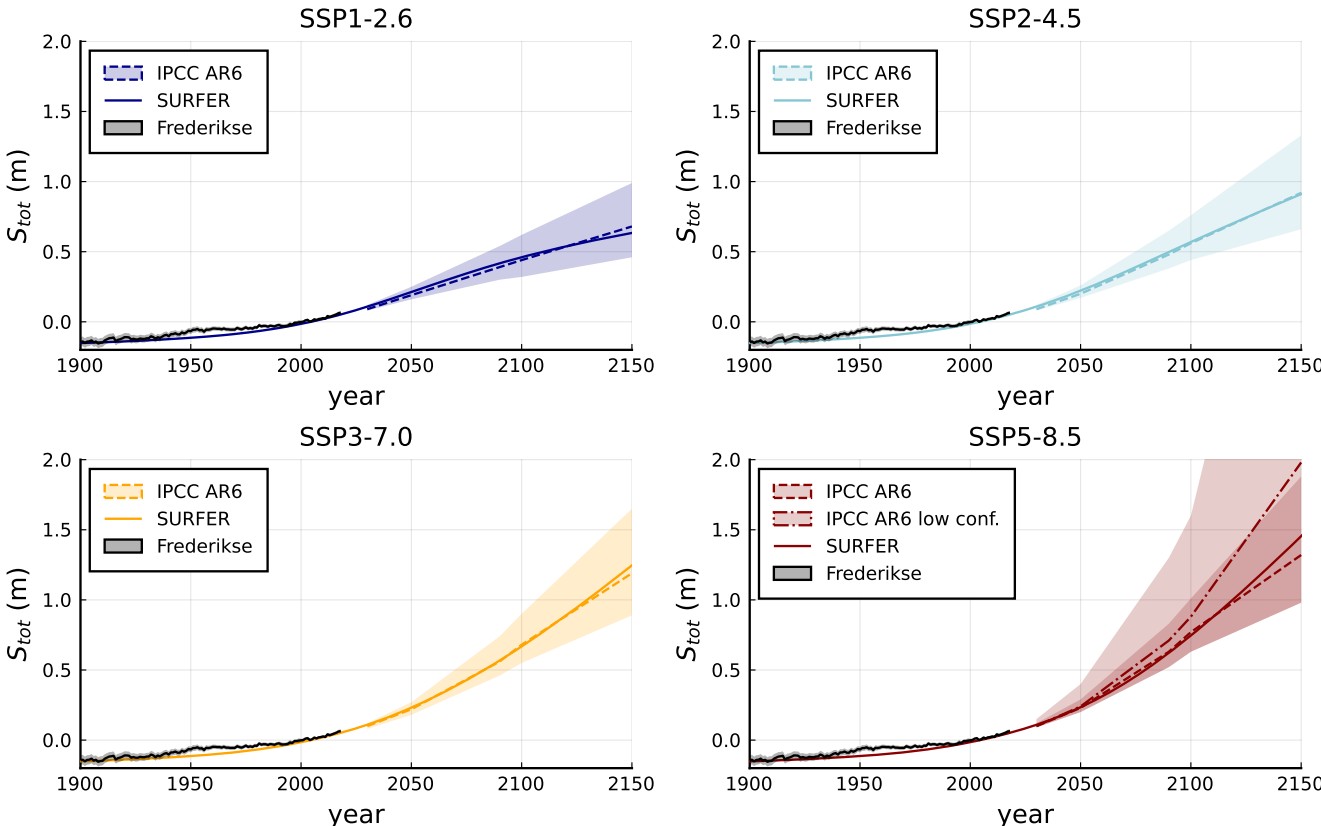

**Figure 14.** Historic and shorter term sea level rise projections. Solid coloured lines show SURFER's total sea level rise estimates. Solid black lines with shaded envelope correspond to historic observations reported by Frederikse et al. (2020), dashed coloured lines correspond to IPCC AR6 estimates and shaded regions to IPCC AR6 likely (medium confidence) ranges. The dash-dot line in SSP5-8.5 corresponds to the low confidence IPCC AR6 estimate. SURFER's and historical results are with respect to year 2004. IPCC results are relative to a baseline of 1995–2014.

the version of the model presented here, there is no feedback from climate (temperature) into the carbon cycle. In Fig. 18 we compare the rate of sulfur injections needed by SURFER and by the ESMs that participated in the GeoMIP, see (Visioni et al., 2021), to accomplish the G6sulfur experiment.

## 4   Discussion

In the literature other reduced complexity models for sea level rise can be found (e.g., Wong et al., 2017; Nauels et al., 2017;
Palmer et al., 2020). Wong et al. (2017) introduce BRICK, a framework for modelling sea level rise. They argue that models for risk analysis should be accessible, transparent, flexible and efficient. In this paper we have shown that SURFER complies with all of these criteria. Additionally, compared to BRICK, SURFER models ocean acidification and incorporates tipping

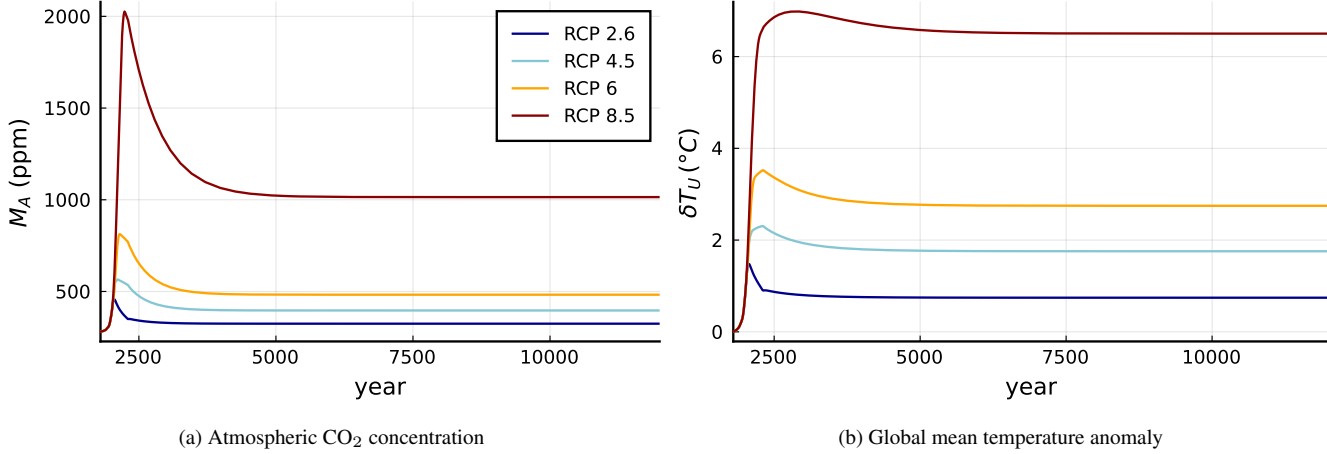

(a) Atmospheric CO$_2$ concentration

(b) Global mean temperature anomaly

**Figure 15.** SURFER's long-term prediction when forced by extended RCP scenarios.

points in the ice sheet components. Such phenomena are important when assessing policies (Lenton and Ciscar, 2013). Nauels et al. (2017) and Palmer et al. (2020) also provide efficient a reduced complexity model and a statistical emulator, respectively, that allow for sea level rise projections. These models, however, are already less transparent than SURFER and again, they do not incorporate tipping point dynamics. Finally, SURFER has been shown to reproduce results from EMICs, ESMs and 3D ice sheet models on timescales from decades to millennia. Is is unclear to us whether the BRICK, MAGICC and the model by Palmer et al. (2020) are applicable on such long timescales. Thus, we are convinced that SURFER is a valuable addition to the literature.

SURFER is simple, easy to understand and fast to run, but it misses some processes, carbon reservoirs and feedbacks. This gives room for extensions, and here we provide relevant information for users who would like to take some of the relevant processes into account, while explaining why we have not included them in the present model.

Permafrost holds approximately 1400 PgC (Canadell et al., 2021): that is about twice as much carbon as currently contained in the atmosphere. The thawing and subsequent release of part of that carbon into the atmosphere may therefore constitute a substantial positive feedback on anthropogenic emissions. Several studies, simulations and observations have been made to quantify the strength of these effects, (MacDougall and Knutti, 2016; Chadburn et al., 2017; Burke et al., 2017; Turetsky et al., 2020; Burke et al., 2020) but the spread in the estimates is considerable. MacDougall and Knutti (2016) made projections of the release of carbon from permafrost soils using a perturbed parameter ensemble with the UVic-ESCM EMIC. Among other things, they computed the sensitivities of cumulative emitted carbon per extra degree of warming. They showed the (transient) sensitivities changed with time, obtaining 24, 39, and 47 PgC/°C for the years 2100, 2200 and 2300 under all RCP scenarios. A similar computation was done by Burke et al. (2017) with the IMOGEN model coupled to two different land models. In that case, although lower values were obtained, the sensitivities remained time dependent ($\sim$ 10, 20, 30 PgC/°C for 2100, 2200 and 2300 under all RCP scenarios). In an observation-based study, Chadburn et al. (2017) estimated an equilibrium sensitivity

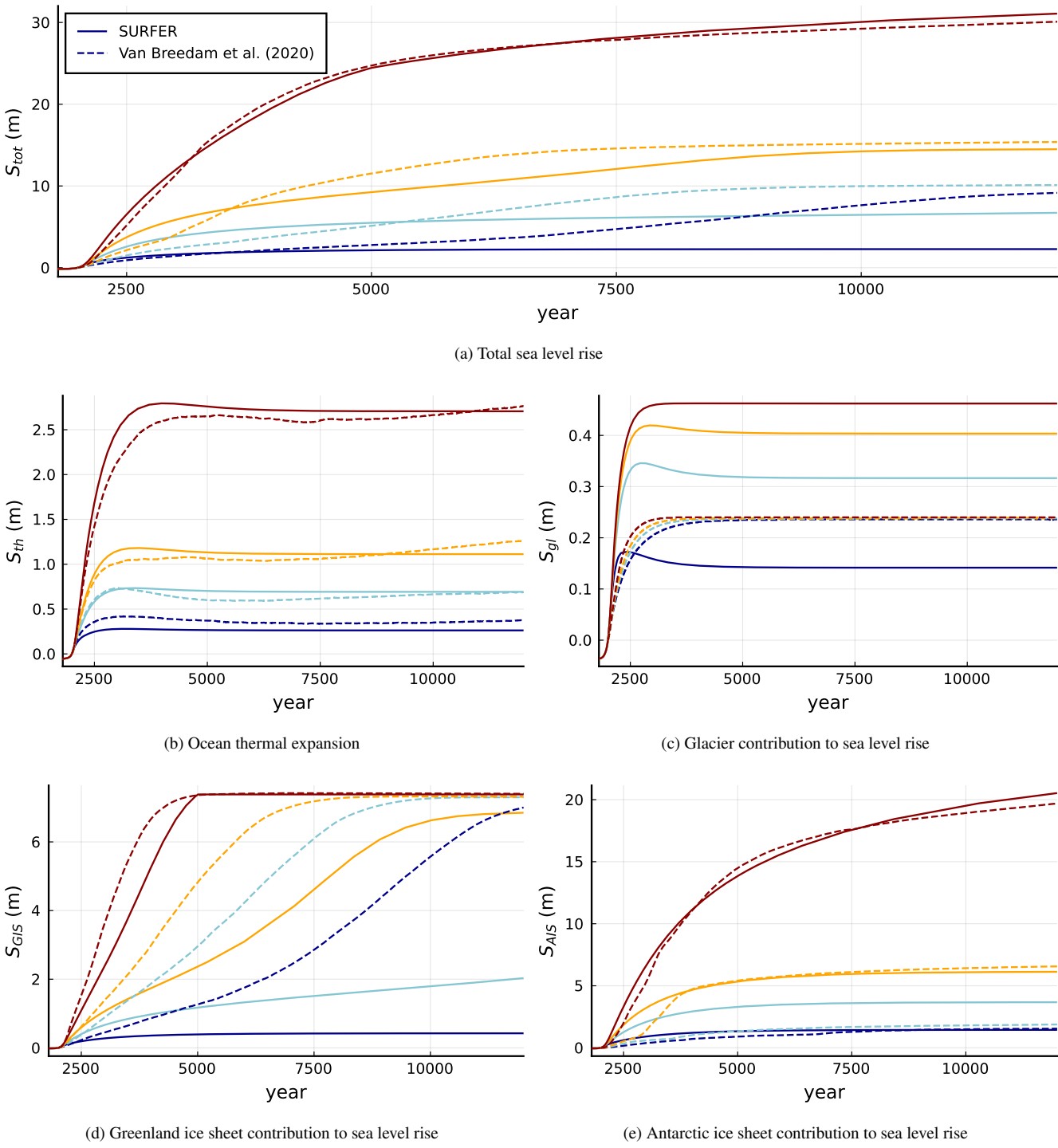

(a) Total sea level rise

(b) Ocean thermal expansion

(c) Glacier contribution to sea level rise

(d) Greenland ice sheet contribution to sea level rise

(e) Antarctic ice sheet contribution to sea level rise

**Figure 16.** Long-term sea level rise from SURFER and Van Breedam et al. (2020) when forced by extended RCP scenarios. Solid lines correspond to SURFER and dashed to the results in (Van Breedam et al., 2020). Colours indicate different RCP scenarios as in Fig. 15.

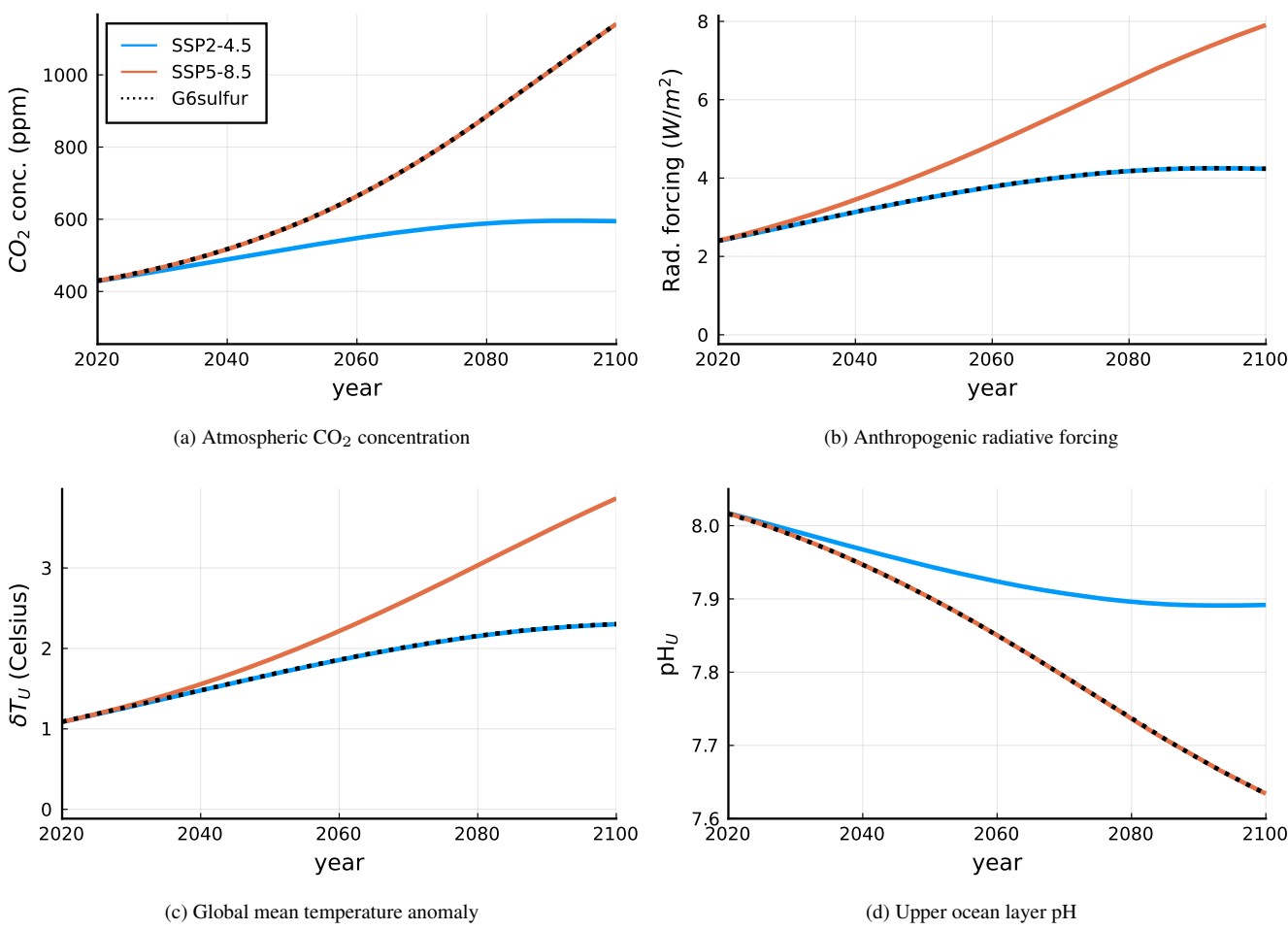

**Figure 17.** SURFER's output for the G6sulfur experiment in black dotted line. Orange and blue lines are the SSP5-8.5 and SSP2-4.5 emission scenarios for comparison.

of permafrost area loss per degree of future global warming of $4.0^{+1.0}_{-1.1}$ million km$^2$/°C. The climatic consequences for CO$_2$
emissions from thawed permafrost were not explored. In the ZECMIP experiments (Jones et al., 2019; MacDougall et al.,
2020), only two of the models (NorESM2-LM and UVicESCM2.10) had a permafrost module. The presence of the module in
these models is visible on the absolute size of their land carbon reservoir compared to the other models (see Figs. 9 and 10),
but not on the evolution of the land reservoir anomalies, suggesting that, in these models at least, the permafrost feedback does
not dominate other carbon fluxes. Burke et al. (2020) acknowledges that modelling future permafrost thaw and its resulting
CO$_2$ emissions remains a challenge to ESMs and EMICs, and points to deeper soils and the inclusion of a representation of
abrupt thawing events as the main points to be improved. For this reason we left the addition of this reservoir and its possible
feedbacks for the future.

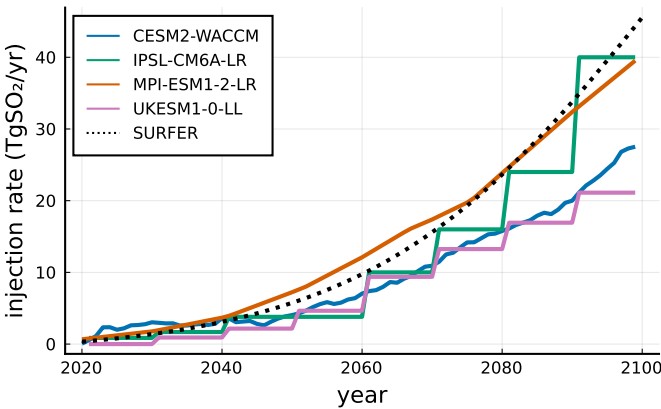

**Figure 18.** Comparison of $SO_2$ injection rate needed to do the G6sulfur experiment by SURFER and by ESMs participating in GeoMIP reported in (Visioni et al., 2021).

The value of the equilibrium climate sensitivity used in SURFER is compatible with that obtained by ESMs that include the effect of ice-albedo feedback from the melting of sea-ice. Consequently, the ice-albedo feedback coming from sea-ice can be thought of as already included in SURFER even if sea-ice is not explicitly part of the model. A representation of the ice-albedo feedback due to the melting of the ice sheets, however, is missing in SURFER. Wunderling et al. (2020) studied the impact of the melting of several cryosphere elements on the global mean temperature. They found that the full melting of Greenland's ice sheet would lead to a temperature increase of $0.13°C$. Approximately 50% of which corresponds to albedo, and the rest to changes in lapse rate, water vapour and clouds. For the west Antarctic ice sheet an increase in temperature of $0.05°C$ was found, again with 50% of it coming from albedo effects. The East Antarctic ice sheet was not included in the study. Adding these effects has been left for future work in which more data from geographically-explicit models becomes available.

SURFER's carbon cycle does not take into account the soft tissue or the carbonate pump which act to create a Dissolved Inorganic Carbon gradient in the water column with the deeper layer containing more DIC than the upper layer. We have introduced a constant parameter, $\delta_{DIC}$, which allows for this difference in DIC between layers to be captured. The soft tissue and carbonate pumps might be affected by temperature and $CO_2$ concentration changes, leading to a time evolving $\delta_{DIC}$, while in SURFER this parameter is constant.

The feedback of temperature on the physical component of the carbon cycle can be included by substituting the solubility constant $K_0$ and the dissociation constants $K_1$ and $K_2$ by their temperature dependent expressions as done in (Glotter et al., 2014). This feedback, which acts in the direction of increasing atmospheric $CO_2$ by reducing the ocean carbon uptake, is however known to be small (Glotter et al., 2014) due to two competing processes; while solubility decreases with temperature, the dissociation constants increase.

Calcium carbonate compensation, a slower process in which the ocean's pH is neutralised by the dissolution of $CaCO_3$ from sediments is not present in SURFER. Calcium carbonate compensation acts on timescales of 3-7 kyr (Archer et al., 2009) and provides an extra buffer for atmospheric $CO_2$. Since the impact from ocean acidification on marine ecosystems takes place

at much shorter timescales, and since the highest and most harmful rates of sea level rise occur before the year 4000 for the RCP scenarios, it seems unnecessary for our purposes to add this effect (together with a sediments reservoir) into SURFER. Due to this choice its predictions will start to deviate from expected on timescales of several thousands of years and ocean acidification will be slightly intensified with respect to models which do include such processes. We have also ignored even longer timescale effects related to the $CO_2$ chemical reaction with silicate rocks.

In SURFER solar radiation modification acts by construction only on the globally averaged temperature. Cao and Jiang (2017) studied the feedbacks between solar radiation modification and the carbon cycle. They found that, under solar radiation modification, the land carbon reservoir becomes more efficient in absorbing $CO_2$. This reduces the total atmospheric $CO_2$ concentration and thus the total amount of $SO_2$ injections needed to reach a certain temperature goal. It also reduces very slightly ocean acidification. On the other hand, Tjiputra et al. (2016) reported an increased ocean carbon uptake in the presence 615 of solar radiation modification and a negligible global mean change in land carbon uptake. While the results by Tjiputra et al. (2016) and Cao and Jiang (2017) show the importance of investigating the subject further, we decided to wait for a more coherent picture before including such feedbacks in SURFER. Such feedbacks could potentially be included by making the land carbon equation, and the solubility and dissociation constants, temperature dependent. The fact that in SURFER solar radiation modification only affects the temperature may implicitly cause an under-appreciation of the risks associated with this 620 technology, especially its direct impact on atmospheric chemistry, circulation, precipitation and its indirect impact on health, food security and ecological systems (Zarnetske et al., 2021). As already said in the introduction, this inclusion is motivated by the need to put in a single coherent framework the contrasting timescales of the (short) residence time of stratospheric aerosols in the atmosphere, the (long) residence time of carbon in the ocean-atmosphere system and the different timescales involved in the sea level response. We nevertheless urge potential users of SURFER interested in evaluating the cost and benefits and 625 related dilemmas associated with solar radiation modification to adequately incorporate its potentially severe adverse side effects in their studies and conclusions. Solar radiation modification impacts cannot be reduced to temperature and any study considering this technology as an option should account for the multifaceted risks associated with it (Robock, 2016, 2020; Zarnetske et al., 2021). Not doing so will likely lead to bad and dangerous advice.

## 5 Conclusions

In this paper we have presented SURFER, a new model that is easy to understand and modify, fast to apply, and hence well-suited to be used for policy assessment. SURFER emulates the results of EMICs and ESMs regarding $CO_2$ concentration, temperature anomalies, ocean acidification and sea level rise. Since it emulates well a range of aggregated quantities, it can be used in policy assessments that value policies according to a wider criteria than just global mean temperature. Furthermore, due to its lightness and ability to correctly represent millennial timescales, it is also well suited for long-sighted decision problems 635 and for commitment and responsibility analyses in presence of uncertainty, among other kinds of policy assessments.

We have shown that SURFER's sea level rise module performs exceptionally well on the short timescales. Moreover, this module includes ice sheet tipping points, which are particularly important for good performance on the long timescales. Being

parametrised on the ice sheet tipping points, the module can easily be updated to match latest research in this fast growing area. With such a flexible ice sheet module, one may envision representing uncertainty about tipping behaviour as parameter uncertainty and investigating the effect of this non-deterministic setting on policy assessments.

Finally, we have shown that SURFER works well under a variety of forcing scenarios. These scenarios do not only include different rates of positive $CO_2$ emissions and a range of total cumulative emissions, but also future technologies such as solar radiation modification and, for SSP1-2.6 and RCP2.6, carbon dioxide removal (in the form of atmospheric negative $CO_2$ emissions). As a consequence, SURFER is well-suited for policy assessments that require considering a variety of forcing scenarios. This is the case for sequential decision problems and also for commitment assessments that capture uncertainty about available future options regarding earth management. This last application will be the main focus of a companion paper that relies on SURFER as it's main computational engine.

*Code availability.* The exact version of SURFER used to produce the results used in this paper is archived on Zenodo (Martínez Montero et al., 2022) under MIT license, as is the input data to run the model and produce the plots for all the simulations presented in this paper. The data corresponding to the results of other references can be found through the corresponding cited references or by personal contact with the authors in the case of the results in (Robinson et al., 2012; Van Breedam et al., 2020; Visioni et al., 2021).

*Author contributions.* Based on pertinent roles defined by CASRAI's CRediT – Contributor Roles Taxonomy: Conceptualization (MM, MC, NB, NB), Methodology (MM, MC, VC), Project Administration (MC), Software (MM), Supervision (MC), Validation (MM), Visualization (MM, NuriaB), Writing – original draft (MM), Writing – review & editing (MC, NB, NB, VC).

*Competing interests.* None

*Acknowledgements.* The authors thank the editors and reviewers, whose comments have lead to significant improvements of the original manuscript. The authors also thank Jonas van Breedam, Alex Robinson and Daniele Visioni for sharing their data. This project is TiPES contribution #146: This project has received funding from the European Union's Horizon 2020 research and innovation programme under grant agreement No 820970". MC is funded as Research Director by the Belgian National Fund of Scientific Research. VC is funded as Research Fellow by the Belgian National Fund of Scientific Research (F.S.R.-FNRS)".

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
