# Peer review of "SURFER v2.0: A flexible and simple model linking anthropogenic $CO_2$ emissions and solar radiation modification to ocean acidification and sea level rise"

_EGUsphere, 2022_

## Referee Comment (RC1)

Martínez Montero et al. present a simple model with the aim to estimate sea level rise from greenhouse gas emissions and solar radiation management. There is clear motivation for such work, but to me two important issues need to be clarified or added before an in-depth review and potential path towards publication:

a) the authors state to model sea level rise in the title and at several points in the manuscript. The model however only captures the ice sheet contributions to sea level rise from Greenland and Antarctica. This is far from the complete picture on sea level rise. This fuzziness is in particular difficult as the authors motivate their work with the aim to improve cost-benefit analysis. In such analysis the total impacts of sea level rise need to be captured, which need to incorporate all major contributors to sea level rise.

b) Reading from the abstract and introduction, sea level rise is a key output metric of the model. This is in contrast to the very short section 2.4.3 on the calibration of the ice sheet components. There is no figure that allows the reader to grasp how well the model performs in comparison to the datasets it aims to emulate. This section needs more details and graphics so the reader can get a clear picture on the performance of the model. It would also be of interest how it performs with respect to historical period for which direct observations of sea level components are available, for example from Frederikse et al. 2020. I would also like to see a more detailed discussion to set this work in context with similar studies on simple sea level models like Wong et al (2017), Nauels et al. (2017) or Palmer et al (2020).

**Minor comments**

L4 what does accurate mean? be more precise?

L13 "(but critizised …)" please simplify sentence construction

L22 "… than they should be" reads like your personal judgement. what do you mean exactly?

L30 " commit future generations …" here the works of Nauels and Mengel may be interesting to cite:

https://www.nature.com/articles/s41467-018-02985-8

https://www.pnas.org/doi/full/10.1073/pnas.1907461116

L333 The Pattyn and Huybrecht models are largely different from PISM underlying Garbe et al. 2020. It seems a zoo of different sources to calibrate your parameters. How do you justify this?

L382ff please extend and make this easier to grasp for the reader. As now the reader does not know from the manuscript how well your model performs.

Intro in general:

You largely motivate your work through a critique on short-sighted solar radiation management studies. I would suggest to broaden this motivation. At least I would extend a bit the "sea level commitment" point as this is a key issue in the climate policy to impacts relation.

**References**

Frederikse, Thomas, et al. "The causes of sea-level rise since 1900." *Nature* 584.7821 (2020): 393-397.

Nauels, Alexander, et al. "Synthesizing long-term sea level rise projections–the MAGICC sea level model v2. 0." *Geoscientific Model Development* 10.6 (2017): 2495-2524.

Palmer, M. D., et al. "Exploring the drivers of global and local sea-level change over the 21st century and beyond." *Earth's Future* 8.9 (2020): e2019EF001413.

Wong, Tony E., et al. "BRICK v0. 2, a simple, accessible, and transparent model framework for climate and regional sea-level projections." *Geoscientific Model Development* 10.7 (2017): 2741-2760.

---

## Author Response (AR1)

**Author response to referees' comments**

August 2, 2022

**AR to RC1**

**Comment 1**  Martínez Montero et al. present a simple model with the aim to estimate sea level rise from greenhouse gas emissions and solar radiation management. There is clear motivation for such work, but to me two important issues need to be clarified or added before an in-depth review and potential path towards publication:

a) the authors state to model sea level rise in the title and at several points in the manuscript. The model however only captures the ice sheet contributions to sea level rise from Greenland and Antarctica. This is far from the complete picture on sea level rise. This fuzziness is in particular difficult as the authors motivate their work with the aim to improve cost-benefit analysis. In such analysis the total impacts of sea level rise need to be captured, which need to incorporate all major contributors to sea level rise.

**Reply to comment 1**  We agree with the reviewer that estimates of sea-level rise need to incorporate all major contributors and that the thermal expansion of the oceans and the melting of glaciers are important ones, especially on timescales of centuries. We have, therefore, added those contributions to the model. We have renamed Sec. 2.3 to "Sea level rise model"; the section now contains three subsections for the considered contributions: ocean thermal expansion, glaciers, and the two ice sheets. The extension, presented in Secs. 2.3.1 and 2.3.2, suggests that the original model can indeed be modified very easily, as argued in the revised introduction. Results for the extended model are discussed in Sec. 3.

**Comment 2**  b) Reading from the abstract and introduction, sea level rise is a key output metric of the model. This is in contrast to the very short section 2.4.3 on the calibration of the ice sheet components. There is no figure that allows the reader to grasp how well the model performs in comparison to the datasets it aims to emulate. This section needs more details and graphics so the reader can get a clear picture on the performance of the model. It would also be of interest how it performs with respect to historical period for which direct observations of sea level components are available, for example from Frederikse et al. 2020. I would also like to see a more detailed discussion

to set this work in context with similar studies on simple sea level models like Wong et al (2017), Nauels et al. (2017) or Palmer et al (2020).

**Reply to comment 2**   We have expanded Sec. 2.4 to explain more clearly the calibration of all sea level rise components. Graphics showing the performance of the sea level rise model against other models have been added in Secs. 2.4 and 3. There we compare SURFER's output on the short timescales (centuries) to the results reported in IPCC AR6, to the projections by the BRICK model [1, 2] and also to the historical observations reported in [3]. We also contrast SURFER's prediction on the timescales of millennia to the results in [4]. We have moreover explained in Sec. 2.4.5 that the existence of a model inter-comparison project of sea level rise on millennial timescales would make calibration and comparison easier and richer. We have extended Sec. 4 to include a discussion to put in context SURFER with respect to the other reduced complexity models to estimate sea level rise mentioned by the referee.

**Minor comments**

- L4 what does accurate mean? be more precise?

  We have rephrased the abstract and the referred sentence is no longer there. We emphasise that SURFER is primarily meant to be a fast, understandable and adaptable tool that captures approximately certain important features of higher dimensional models. Further clarification has been added in the introduction.

- L13 "(but critizised ...)" please simplify sentence construction

  We have substantially rewritten the introduction and that sentence is not there anymore.

- L22 "... than they should be" reads like your personal judgement. what do you mean exactly?

  We have substantially rewritten the introduction and that sentence is not there anymore.

- L30 " commit future generations ..." here the works of Nauels and Mengel may be interesting to cite:
  https://www.nature.com/articles/s41467-018-02985-8
  https://www.pnas.org/doi/full/10.1073/pnas.1907461116

  We have cited the suggested references in the introduction, see lines 19-20.

- L333 The Pattyn and Huybrecht models are largely different from PISM underlying Garbe et al. 2020. It seems a zoo of different sources to calibrate your parameters. How do you justify this?

> To calibrate SURFER's ice sheet model we ideally need 3 things: steady state structure, dynamic melting experiments and dynamic accumulation experiments. In general, the three experiments have not been performed by the same model. That is why we have to rely on such a zoo of different sources for the calibration. We have justified in more detail the usage of the different sources in Section 2.4.5.

- L382ff please extend and make this easier to grasp for the reader. As now the reader does not know from the manuscript how well your model performs.

  > We have added Figs. 5, 6, 7 and 16 comparing sea level rise output of SURFER to other models' output. The discussion of these figures clarifies how well the model performs.

**Intro in general**   You largely motivate your work through a critique on short-sighted solar radiation management studies. I would suggest to broaden this motivation. At least I would extend a bit the "sea level commitment" point as this is a key issue in the climate policy to impacts relation.

**Reply to Intro in general**   We fully agree with the reviewer and we have substantially rewritten the introduction to better motivate the the design choices in the development and the intended usages of SURFER. We start the introduction discussing the importance of taking into account sea level commitment and ocean acidification when assessing $CO_2$ emission policies. We also mention that studying sea level commitment is one of the intended usages that will be presented in a companion paper.

**AR to RC2**

This paper presents a new simple and fast modelling tool (SURFER) that allow assessing carbon emission evolution and solar radiation management long-term impacts on sea-level rise and ocean acidification. The paper describes the tool, presents example of calibration and associated model results for a few set of experiments. Although I find this work interesting and I believe that SURFER could indeed be useful for the community, this paper does not really demonstrate why SURFER should be used and, in my opinion, somehow lacks of maturity in its present version. Please find below few suggestions/comments that I believe may help improving the manuscript.

**Main comments**

**Framing and motivation**

- I think it would be good to make clearer what SURFER fills as a gap; because EMICs and ESMs can also be used for long runs (as shown in the paper actually), but I understand that they don't allow estimating SLR and ocean acidification. Is

that right?

EMICs and ESMs can also be used for long runs and some also estimate sea level rise and ocean acidification. However, for some policy assessment applications, as sequential decision problems and responsibility under uncertainty, among others, it is impractical to use them. SURFER is well suited for those applications because it is:

- Fast to apply: as detailed in Sec. 2.5
- Easily understandable: all the model details, equations and parameters needed to re-code the model are contained in Sec. 2. Additionally, the provided code is easy to read and use.
- Easily modifiable: in a few of weeks, since the referees posted their comments, we have been able to extend the sea level rise module to include sea level rise contribution from glaciers and sea level rise.

In the introduction we now explain the gap filled by SURFER and elaborate on the design features of SURFER that we consider necessary for the mentioned applications.

- Most of the introduction moves around the fact that such simple models are useful to explore impact of policy analyses, cost-benefit analyses, etc... for long-term concerns that are mostly omitted. These omissions may in turn lead to wrong decisions. In the conclusion, one can read "[...] it is well-suited for long-sighted multi-objective policy analyses: With this model we can not only assess the short and long-term effects of anthropogenic emissions, but also put future technologies into the mix, such as carbon dioxide removal and solar radiation management." Although I find such motivation for developing that sort of model indeed appropriate, I don't believe that results shown in the paper demonstrate this. My suggestion would thus to really perform some policy analyses-like case studies to make this more concrete and support the motivation of the paper. Otherwise, the introduction should rather focus on what the model provides as outputs and how this fills more direct needs.

Indeed we do not demonstrate or provide an example of SURFER being used for the mentioned applications in this manuscript. As now explained in the introduction this would go well beyond the scope of this paper and we have decided to do so in dedicated companion papers. We have, as suggested by the referee, shifted the focus of the introduction to what features the model provides and why those features are important for its intended usage.

- Ocean acidification is another very important output that should be better emphasized in the title too, not just SLR.

We have changed the title accordingly.

**Model design & description**

- The model provides SLR estimates due to Greenland and Antarctic melting components but omit first order components such as glaciers, ice caps and thermal expansion. I'd suggest either to upgrade the model to allow accounting for those non-negligible contributions or otherwise to clarify that the model provides "SLR due to ice-sheets melting".

  We have added the additional sea level rise contributions to the model. We have renamed Sec. 2.3 as "Sea level rise model" which contains three subsections for the considered contributions: ocean thermal expansion, glaciers, and the two ice sheets.

- In the model, the carbon cycle processes that rule carbon fluxes between the atmosphere and ocean rely on chemical & physical processes that are well known and constrained. For the land reservoir though, this appears to be derived from EMICs and ESM simulations and ends up to be model dependent. In this regard, I think it would be good to estimate or comment on how sensitive are the SURFER results to such a parameterisation. Is there no other way or can we put additional constraints to lower the influence of external model results?

  As now clarified in the introduction, SURFER is not meant to be a "better" model than the reference ESMs it is intended to emulate. Rather, SURFER is a tool that trades model complexity for speed and understandability. The proposed parameterisation for the land reservoir equation serves this purpose. An explanation for the interactions responsible for the form of the parameterisation has been added in Sec. 2.1.3. The proposed parameterisation also allows for an easy calibration as new model results become available.

- In my opinion, the ice-sheet module would deserve some more explanation on how the equations on ice volume are derived and to what the different terms refer to – note that I have basically no knowledge on ice-sheet modelling and physics. Details could be provided in a supplementary material.

  Further clarification has been added to Sec 2.3.3. Section 2.4.5 has been expanded and hopefully helps to clarify the main idea behind the ice-sheet module.

**Model validation & comparison**

- Section 3 shows some qualitative comparison with CMIP5 and CMIP6 results for some variables but in my opinion, it would be worth to perform a more systematic comparison with CMIP5 & CMIP6 outputs. These data that are easily accessible. Such an analysis would make the models comparison and SURFER performances assessment more convincing.

In assessing the performance of the carbon cycle and climate sub-models of SURFER, we have taken advantage of the ZECMIP dataset and of the pH plot from CMIP5 and CMIP6 of Ref. [5]. CMIP5 and CMIP6 data are typically provided on 2D or 3D grids and we are not aware of other consistently reduced 0D CMIP5-6 data sets beyond those mentioned above. We would be grateful if you could provide us pointers to such data and we would be glad to present a more systematic comparison with CMIP5-6 outputs. In our revision, we have extended and improved the comparison of the sea level rise contributions both on short and long timescales. This has been done by comparing the output of SURFER to the output of BRICK [1, 2] (another reduced model), to the data provided in the IPCC AR6 WG1 Chapter 9, to the historical observations of Ref. [3] and to the results provided in Ref. [4].

- The Greenland ice-sheet contribution to SLR strongly departs from the other study to which the results are compared (Van Breedam et al., 2020). This suggests a strong sensitivity of the model results to the calibration phase. In this regard, I think that some illustration of the sensitivity to some parameters (e.g. from Table 3 or Table 4) could be very relevant. Moreover, this appears to be easily achievable because the model is particularly fast. From a more general point of view, I think the strength of the SURFER model is that it is very efficient to run long period. One could imagine using such a tool to propagate uncertainty on parameters or boundary conditions.

  We have expanded Sec. 2.4.5. There we now explain that the lack of consistent millennial timescale experiments for different ice sheet models leads us to calibrate SURFER, except for the accumulation timescale, against only one model in the case of Greenland, that is the one in Ref. [6]. We have expanded the explanation on the discrepancy of SURFER's Greenland results to those of Ref. [4] and also argued why the results of Ref. [6] are better suited to calibrate SURFER. Once a model inter-comparison project for ice sheets on millennial timescales is available it would be very interesting to encode inter-model uncertainty as parameter uncertainty in SURFER.

**Specific comments**

- L201: Planetary boundary → this is an important point and could be very use as a motivation in the introduction I believe.

  Thanks! We have implemented your recommendation and mentioned how SURFER could be applied to take into account the planetary boundaries in the values of different policies, lines 15-16, or to define the safe operating spaces of [7] in terms of planetary boundaries in lines 33-34 of the revised introduction.

- Section 2.2: providing units would be useful to better follow the equations. Eq 32: it would be good to describe all terms in the text below.

All units of parameters are provided in Table 3, we think that adding them in the main text would make reading more complicated. We have clarified the units of the state variables, ECS and anthropogenic forcing in the main text.

- Section 2.3: see the main comments, but more details in a supplementary material would be welcome here.

  We have added substantially more details in the main text in Sec. 2.3 and 2.4 and 3.

- L289: why reasonable value? compared to what ? Please clarify.

  It's a reasonable value as compared with global average data in Fig. 8.1.2 of Ref. [8] and to global averages from ZECMIP data, see pre-industrial range of values in DIC plot in Fig. 9 of the manuscript. We have added the clarification in the text.

**References**

[1] Tony E Wong, Alexander M R Bakker, Kelsey Ruckert, Patrick Applegate, Aimée B A Slangen, and Klaus Keller. BRICK v0.2, a simple, accessible, and transparent model framework for climate and regional sea-level projections. *Geoscientific Model Development*, 10(7):2741–2760, jul 2017.

[2] Alexander M.R. Bakker, Tony E. Wong, Kelsey L. Ruckert, and Klaus Keller. Sea-level projections representing the deeply uncertain contribution of the West Antarctic ice sheet. *Scientific Reports*, 7(1):1–7, 2017.

[3] Thomas Frederikse, Felix Landerer, Lambert Caron, Surendra Adhikari, David Parkes, Vincent W. Humphrey, Sönke Dangendorf, Peter Hogarth, Laure Zanna, Lijing Cheng, and Yun Hao Wu. The causes of sea-level rise since 1900. *Nature*, 584(7821):393–397, 2020.

[4] Jonas Van Breedam, Heiko Goelzer, and Philippe Huybrechts. Semi-equilibrated global sea-level change projections for the next 10 000 years. *Earth System Dynamics*, 11(4):953–976, 2020.

[5] Lester Kwiatkowski, Olivier Torres, Laurent Bopp, Olivier Aumont, Matthew Chamberlain, James Christian, John Dunne, Marion Gehlen, Tatiana Ilyina, Jasmin John, Andrew Lenton, Hongmei Li, Nicole Lovenduski, James Orr, Julien Palmieri, Jörg Schwinger, Roland Séférian, Charles Stock, Alessandro Tagliabue, Yohei Takano, Jerry Tjiputra, Katsuya Toyama, Hiroyuki Tsujino, Michio Watanabe, Akitomo Yamamoto, Andrew Yool, and Tilo Ziehn. Twenty-first century ocean warming, acidification, deoxygenation, and upper ocean nutrient decline from CMIP6 model projections. *Biogeosciences Discussions*, pages 1–43, 2020.

[6] Alexander Robinson, Reinhard Calov, and Andrey Ganopolski. Multistability and critical thresholds of the Greenland ice sheet. *Nature Climate Change*, 2(6):429–432, 2012.

[7] J. Heitzig, T. Kittel, J. F. Donges, and N. Molkenthin. Topology of sustainable management of dynamical systems with desirable states: from defining planetary boundaries to safe operating spaces in the Earth system. *Earth System Dynamics*, 7(1):21–50, jan 2016.

[8] J.L. Sarmiento and N. Gruber. *Ocean Biogeochemical Dynamics*. Princeton University Press, 2006.

---

## Author Response (AR2)

**Author response to comments from referee #3**

September 30, 2022

**Comments and replies**

I've been asked to review the revised version of the manuscript for the first time. The aim of this paper is to describe a simple model linking CO2 emissions to ocean acidification and sea level rise; numerous updates have been suggested already in the first round of review.

I'm not an expert on ice sheet modeling, but I think the first 2 reviewers addressed that part already and the authors followed all of the suggestions and improved the manuscript compared to the fist iteration.

In general, I think that such a fast, simple model would be a pretty interesting addition to the literature. However, the results that the authors obtain for SRM have been already discussed in the literature before: for instance (Zarnetske et al. 2021) talk about the fact that ocean acidification wouldn't be reduced, and while the land carbon sink might be increased (see for instance Cao and Jiang, 2017), that's really not by much. It would be good for the authors to acknowledge the work done in this area and say that their results confirm (or not if they don't) previous findings.

Thank you for the positive comments and suggestions. We have taken them into account in the following way. Regarding the SRM results, we have explained how SRM in SURFER only affects the temperature by construction, i.e., there are no feedbacks into the carbon cycle (ll. 540 - 544). We have also expanded the discussion on SRM (ll. 607 - 625), citing both of the suggested references and including a discussion on how feedbacks into the carbon cycle could be added to a future release of the model.

**A few general comments**

- The paper also deals with SRM but the title does not suggest that: "emissions" is a pretty vague term to include both anthropogenic CO2 emissions and SRM. I suggest to change it to something that actually explains it's about both.

We see your point and we have changed the title to: "SURFER v2.0: A flexible and simple model linking anthropogenic CO2 emissions and solar radiation modification to ocean acidification and sea level rise".

- By the way, the term that is now more widely used for SRM is Solar Radiation Modification and not Management (see NASEM report in 2021) or Sunlight Reflection Methods.

  Thank you for pointing this out. We have changed the term to Solar Radiation Modification throughout the manuscript.

**Introduction**

- I would suggest at least explaining what SRM is in the introduction - it might not be so widely known as the authors think and a reader can't be expected to go look at the references provided.

  We added an explanation of solar radiation modification in the introduction (ll. 13 - 17).

- L 12: because of the long

  We have fixed this typo. Thanks.

- L 14: not sure "value" is the right word here: assessing the efficacy is perhaps a better term

  By assessing the value we actually mean to assign a value judgement in which efficacy might not be the sole criterion. To clarify we have changed the expression to "assign a value judgement".

**Section 2**

- L 79: SO2 is a gas, so "SO2 aerosol injections" is not really a meaningful definition.

  Indeed. We have modified this to "$SO_2$ injections" here and throughout the manuscript.

- L 412: I would remove the square brackets [] for the estimates of ECS and TCR.

  We have removed the square brackets.

**Section 3**

- L 480: " We have considered that the Solar Radiation Management sulphur injections remain zero during the whole simulation period." unclear of why it is relevant here. The RCPs and SSPs do not have SRM, and the G6 experiment is another thing and discussed in another section. This phrase is repeated multiple times in the following paragraphs and it really doesn't need to: maybe say at the

beginning of Section 3 when you'll be considering SRM instead of all the scenarios where you're not.

Thank you for the suggestion. In the beginning of Sec. 3 we have added an overview of the different examples presented and clarified there that only the example of Sec. 3.5 considers SRM (ll. 466 - 470).

- Figure 9 (and following figures): it is customary to name the panels in alphabetical order to make it easy on the reader; the legend hides one of the panel's results.

  We have reduced the legend fontsize so that it does not hide the results in one of the plots in Figs. 9 and 10. For the panels we have included the different plots as subfigures each with its own subcaption.

- Fig. 18: The comparison between SURFER results and G6 is interesting, but it could be done in just one plot. If the authors want to contact me (being the author of the 2021 paper cited, dv224@cornell.edu) I'll gladly give them the SO2 values to combine the two figures in one.

  Thank you for sharing the data with us. We have combined the two plots into a single one, see Fig. 18, and we have thanked you in the acknowledgements.